# Sensitivity of the tropical Atlantic to vertical mixing in two ocean models (ICON-O v2.6.6 and FESOM v2.5)

Swantje Bastin[1], Aleksei Koldunov[2], Florian Schütte[4], Oliver Gutjahr[1], Marta Agnieszka Mrozowska[3], Tim Fischer[4], Radomyra Shevchenko[1], Arjun Kumar[1], Nikolay Koldunov[2], Helmuth Haak[1], Nils Brüggemann[1], Rebecca Hummels[4], Mia Sophie Specht[1], Johann Jungclaus[1], Sergey Danilov[2], Marcus Dengler[4], and Markus Jochum[3]

[1]Max Planck Institute for Meteorology, Hamburg, Germany
[2]Alfred Wegener Institute, Bremerhaven, Germany
[3]Niels Bohr Institute, University of Copenhagen, Copenhagen, Denmark
[4]GEOMAR Helmholtz Centre for Ocean Research Kiel, Kiel, Germany

**Correspondence:** Swantje Bastin (swantje.bastin@mpimet.mpg.de)

**Abstract.** Ocean General Circulation Models still have large upper-ocean biases, including in tropical sea surface temperature, that are possibly connected to the representation of vertical mixing. In earlier studies, the ocean vertical mixing parameterisation has usually been tuned for a specific site or only within a specific model. We present here a systematic comparison of the effects of changes in the vertical mixing scheme in two different global ocean models, ICON-O and FESOM, run at

a horizontal resolution of 10 km in the tropical Atlantic. We test two commonly used vertical mixing schemes; the K-Profile Parameterisation (KPP) and the Turbulent Kinetic Energy (TKE) scheme. Additionally, we vary tuning parameters in both schemes, and test the addition of Langmuir turbulence in the TKE scheme. We show that the biases of mean sea surface temperature, subsurface temperature, subsurface currents and mixed layer depth differ more between the two models than between runs with different mixing scheme settings within each model. For ICON-O, there is a larger difference between TKE and

KPP than for FESOM. In both models, varying the tuning parameters hardly affects the pattern and magnitude of the mean state biases. For the representation of smaller scale variability like the diurnal cycle or inertial waves, the choice of the mixing scheme can matter: the diurnally enhanced penetration of equatorial turbulence below the mixed layer is only simulated with TKE, not with KPP. However, tuning of the parameters within the mixing schemes does not lead to large improvements for these processes. We conclude that a substantial part of the upper ocean tropical Atlantic biases is not sensitive to details of the

vertical mixing scheme.

## 1 Introduction

The sea surface temperature (SST) in the tropics has a major influence on both the local and global atmospheric circulation and climate. Because it affects the location and strength of atmospheric convection, it influences large-scale tropical wind and precipitation patterns, especially over the surrounding continents (e.g. Rouault et al., 2003; Okumura and Xie, 2004; Kucharski

et al., 2009; Giannini et al., 2004; Crespo et al., 2019). By controlling tropical convection, the tropical Atlantic SST can

also influence extratropical climate via teleconnections (e.g. Sardeshmukh and Hoskins, 1988; Cassou et al., 2005). However, tropical oceans are poorly represented in General Circulation Models (GCMs) (e.g. Toniazzo and Woolnough, 2014; Richter, 2015; Lübbecke et al., 2018; Richter and Tokinaga, 2020). Models usually suffer from a warm SST bias in the eastern tropical Atlantic associated with a too weak and delayed Equatorial Cold Tongue, as well as a cold SST bias in the western tropical oceans, leading to a reversed zonal SST gradient in boreal summer compared to observations (Davey et al., 2002; Richter and Xie, 2008; Richter, 2015; Richter and Tokinaga, 2020). In the atmosphere, the GCMs generally show weaker than observed trade winds, which is strongly coupled to the erroneous SST gradient through the Bjerknes feedback (e.g. Bjerknes, 1969; Keenlyside and Latif, 2007). It has been suggested that the weak trade wind bias and thus also the SST bias mostly originates from the atmospheric component of the GCMs, because the trade wind bias peak appears earlier than the peak of the zonal SST gradient bias (Richter and Xie, 2008). Additionally, atmosphere-only models also have been shown to have too weak Atlantic trade winds (e.g. Zermeño-Diaz and Zhang, 2013; Richter et al., 2014). However, uncoupled ocean general circulation models (OGCMs) generally show similar SST biases as the coupled GCMs (e.g. Song et al., 2015; Tsujino et al., 2020; Zhang et al., 2022). These studies suggest that both the oceanic and atmospheric components contribute to the strong tropical biases in the coupled models, which are then amplified by the Bjerknes feedback. Some studies have shown that tropical biases decrease in coupled models with high atmospheric resolution (e.g. Milinski et al., 2016; Harlaß et al., 2018) and high oceanic resolution (e.g. Seo et al., 2007; Small et al., 2014).

Apart from horizontal resolution, one important oceanic process that controls tropical SST is vertical turbulent mixing (e.g. Jochum and Potemra, 2008; Moum et al., 2013; Hummels et al., 2014, 2020). It affects the vertical temperature distribution by inducing down-gradient temperature (as well as other tracer and momentum) fluxes when turbulence energy is available, thus influencing e.g. the SST, air-sea heat fluxes, the thickness of the surface mixed layer, and the diapycnal heat transport across the bottom of the mixed layer. Vertical turbulent mixing is important e.g. for the seasonal cycle of the tropical Atlantic SST, namely the development of the Atlantic Cold Tongue (ACT). The ACT develops in the eastern equatorial Atlantic in boreal summer, when the trade winds intensify due to the northward shift of the Intertropical Convergence Zone (ITCZ). The stronger trade winds in turn intensify the westward-flowing surface current, which increases the shear with the underlying eastward-flowing Equatorial Undercurrent (EUC). This shear enhances vertical turbulent mixing, which cools the surface mixed layer temperature from below (Hummels et al., 2014; Lübbecke et al., 2018). Vertical turbulent mixing also plays a role for tropical Atlantic variability on smaller than seasonal time scales, for example the diurnal cycle of the near-surface temperature distribution (e.g. Smyth et al., 2013; Moum et al., 2022a).

Vertical turbulent mixing is a subgrid-scale process even at high vertical resolution, and therefore needs to be parameterised in ocean models. Several parameterisations of varying complexity are available, either based on empirical considerations or attempting a statistical closure of the fluctuation correlations of the Reynolds-averaged governing equations (Burchard and Bolding, 2001). Statistical closures, in principle, consist of an infinite number of differential equations which are truncated after the first few equations for computational efficiency (Burchard and Bolding, 2001). OGCMs mostly use first order turbulence schemes because higher-order schemes are too computationally expensive. Frequently used vertical mixing schemes in OGCMs include the Richardson number dependent PP scheme (Pacanowski and Philander, 1981), the empirical K-profile

parameterisation (KPP) scheme (Large et al., 1994), and the TKE scheme, which is a 1.5 level statistical closure (Gaspar et al., 1990).

Because vertical mixing affects the SST and thus air-sea heat exchange, improvements in its representation in ocean models could contribute to a reduction of the long-standing biases in the models' tropical SST and climate. One way to improve vertical mixing in ocean models is to tune parameters that are not well constrained by observations, another way is using a different parameterisation. Li et al. (2001) showed that the KPP scheme led to a better representation of the tropical Pacific than the PP scheme, and Blanke and Delecluse (1993) showed that the TKE scheme performs better than the PP scheme, especially for the simulation of the Equatorial Undercurrents. Concerning parameter tuning, Deppenmeier et al. (2020) showed that increasing the $c_k$ parameter in the TKE scheme leads to reduced SST biases in the tropical oceans, and Zhang et al. (2022) could reduce the tropical Atlantic subsurface warm bias by increasing the interior ocean background diffusivity in the KPP scheme. While each of these studies focused on the effect of or across single aspect of vertical mixing, Gutjahr et al. (2021) did a comprehensive (global) investigation of the differences in a single model between four different vertical mixing schemes: PP, KPP, TKE, and TKE+Idemix. They conclude that the optimal choice of the mixing scheme depends on the region and the variable. In their simulations, the large-scale SST bias was insensitive to changes in the vertical mixing scheme. Since these studies all use a single model, it is unclear to what extent their results are applicable to other models. Moreover, remaining biases in all studies might also be due to model errors other than the vertical mixing scheme. To alleviate this, we use here two different ocean models, FESOM and ICON-O, and perform coordinated sensitivity experiments to compare the performance of two commonly used state-of-the-art OGCM vertical mixing schemes (KPP and TKE), as well as the effect of different parameter choices for each of the two schemes.

We focus on the tropical Atlantic because there are several current observational programs with a focus on tropical Atlantic climate. Data that we use to validate the models include hydrographic measurements from Argo floats (Argo, 2022), data from the Prediction and Research Moored Array in the Tropical Atlantic (PIRATA, Bourlès et al., 2019), and data collected during multiple cruises in the tropical Atlantic.

The remainder of the manuscript is organized as follows. In Section 2, we provide a description of the two ocean models that we use for our study, followed by a description of the KPP and TKE mixing schemes in Section 3. The following sections show the results of our mixing scheme and parameter comparisons. We first assess the effect of the vertical mixing parameterisation and its parameter settings on different large-scale features of the tropical Atlantic upper ocean. Among these are the mean mixed layer depth (Section 4.1), the mean surface and vertical temperature structure as well as the seasonal evolution of the surface temperature, including in the Atlantic cold tongue region (Section 4.2), and the mean equatorial current systems (Section 4.3). In addition to the mean representation of the upper tropical Atlantic, we assess the representation of small-scale variability in the different model runs, including Near-Inertial Waves (Section 5.1), and the upper ocean diurnal cycle (Section 5.2). Finally, to put the sensitivity to mixing into perspective, we investigate the effect of different sets of default atmospheric forcing bulk formulae in ICON-O and FESOM (Section 6).

## 2 Model descriptions

We use two different ocean models, which are both part of the European Union Horizon 2020 NextGEMS project's model development effort: the ocean component (Korn et al., 2022) of the ICON Earth System Model (Jungclaus et al., 2022) and the ocean model FESOM (Danilov et al., 2017; Scholz et al., 2019, 2022), which is coupled to the atmosphere model IFS for NextGEMS. The two ocean models will be described in the following, as well as their setup for this study.

### 2.1 FESOM

FESOM2 is a global unstructured-mesh ocean model developed at the Alfred Wegener Institute, Helmholtz Centre for Polar and Marine Research (AWI) in Bremerhaven (Danilov et al., 2017). It is formulated on a triangular mesh, utilizes a finite-volume dynamical core and Arbitrary Lagrangian–Eulerian (ALE) vertical coordinates (Scholz et al., 2019). The model has a computational performance comparable to structured-mesh models (Koldunov et al., 2019) and its unstructured nature enables different types of local mesh refinements (e.g. one that follows local sea surface height variability, Sein et al., 2017). FESOM2

uses the FESIM sea ice model (Danilov et al., 2015), it uses zero-layer thermodynamics (Semtner, 1976) and includes an elastic-viscous-plastic (EVP) solver. For this study we use a FESOM2 mesh that has 50 km resolution over most of the globe, except for the equatorial Atlantic between 25°S and 25°N, where it is set to 13 km resolution. FESOM uses a $z^*$ vertical coordinate, where the total change in SSH is distributed equally over all layers, except the layer that touches the bottom.

### 2.2 ICON-O

ICON-O (Korn et al., 2022) is the ocean component of the **Ico**sahedral **N**onhydrostatic Weather and Climate Model (ICON) in its Earth System Model configuration (ICON-ESM, Jungclaus et al., 2022). It is developed at the Max Planck Institute for Meteorology (MPI-M) in Hamburg. The ocean component of ICON-ESM, ICON-O, solves the hydrostatic (the "Nonhydrostatic" in the name only refers to the atmospheric component) Boussinesq equations with a free surface. These equations are solved on a triangular horizontal grid, which is generated by dividing the spherical domain into an icosahedron and subsequent

division of the 12 icosahedron parts into triangles. In this study, an ICON grid with globally approximately uniform horizontal resolution of about 10km is used. While the ICON and FESOM horizontal grids are different, we argue that the two models are comparable, as we only compare results from our region of interest, i.e. the tropical Atlantic. In the vertical, a $z^*$ coordinate is used where model levels follow the free surface. For details on the numerics or other specifics about ICON-O, see Korn (2017) and Korn et al. (2022).

### 2.3 Common settings and experiment descriptions

Before running the coordinated sensitivity experiments, we decided on settings that would be shared by both models. Although we tried to homogenise the model settings between FESOM and ICON-O that are directly connected to the representation of vertical mixing, we left the rest of the model settings as they are commonly used in the FESOM and ICON-O communities at our institutes, i.e. partly different between the two models. This is intended and part of the reason why we do this study, to see

how much of the variation between the different vertical mixing settings is model specific and how much happens similarly in both models.

We implemented a common vertical axis with 128 vertical levels for both models, with thicknesses ranging from 2 m near the surface to about 200 m near the seafloor. We also agreed to use the same vertical mixing schemes (TKE and KPP), as well as the same parameter settings. To make sure that our implementations of the TKE and KPP schemes are comparable, we use the versions provided by the CVMix (Community ocean Vertical Mixing) project, which has developed a library of standardised vertical mixing parameterisations to be used in ocean models (Griffies et al., 2015; Van Roekel et al., 2018). In the TKE scheme, we vary the $c_k$ parameter (see Section 3.1, Eq. 2). For the KPP scheme, we run the models once with the default setting of $Ri_{crit} = 0.3$, and once with a reduced value of $Ri_{crit} = 0.27$. We do this because the best value for the critical bulk Richardson number is resolution dependent (e.g. Large et al., 1994). The resolution dependence follows from the bulk Richardson number being an approximation of the exact gradient Richardson number due to the finite thickness of the model levels. As the thickness decreases, the bulk Richardson number converges on the gradient Richardson number. Similarly, the critical bulk Richardson number chosen should converge on the critical gradient Richardson number of 0.25 with decreasing thickness. All model runs with their parameter settings are listed in Table 1.

We force all model runs with hourly ERA5 reanalysis (Hersbach et al., 2020), and run them from the end of the 5-year spinup with adjusted mixing parameter settings for two years (2014 and 2015). Of these, we analyse only the second year when the upper ocean has sufficiently adjusted to the changed mixing settings. The year 2015 was chosen because a particularly strong Near-Inertial Wave (NIW) mixing event occured in that year and was observed during a RV Meteor cruise in the tropical North Atlantic, as described and analysed by Hummels et al. (2020). We compare this unique set of observations against our models to assess whether they can reproduce deep reaching NIW mixing events like the one in 2015.

Unfortunately we cannot run the sensitivity runs longer than two years due to the high horizontal resolution and restricted computing resources. However, the adjustment of the upper ocean to the changes in the mixing parameters should happen on a time scale much less than a year, so that the data from 2015 (the second year of our integrations) should be suited to assess the effect of changing the vertical mixing parameters on upper ocean model performance. We checked how representative the model biases from the year 2015 are compared to other years in a longer similar model simulation. For both models, the interannual variability of the annual mean biases is smaller than the biases themselves, and the large scale bias patterns stay the same.

One notable difference between ICON-O and FESOM is the parameterisation of the surface fluxes, for which different bulk formulae are used for the ERA5 forcing. To investigate the effect of the bulk formulae, we did an additional run with ICON-O using the standard FESOM bulk formulae. The default bulk formulae in ICON-O for the ERA5 forcing are those of Kara et al. (2002) over ocean and sea ice, with water vapor pressure and 2 m specific humidity calculated using the (modified) equations from Buck (1981) and longwave radiation calculated using Berliand (1952). FESOM instead uses bulk formulae calculated according to Large and Yeager (2009) over the ocean and with constant bulk exchange coefficients over sea ice, as described in Tsujino et al. (2018). These are also implemented in ICON-O, but usually only used together with JRA55-do forcing (Tsujino et al., 2018). For more details on ICON-O's standard bulk formulae with ERA5-forcing, see Section A in the Appendix.

**Table 1.** Overview of model runs used in this study

| Experiment | Model | Mixing scheme | Parameter settings |
| --- | --- | --- | --- |
| F_TKE_01 | FESOM | TKE | $c_k = 0.1$ |
| F_TKE_02 | FESOM | TKE | $c_k = 0.2$ |
| F_TKE_03 | FESOM | TKE | $c_k = 0.3$ |
| F_KPP_030 | FESOM | KPP | $Ri_{crit} = 0.3$ |
| F_KPP_027 | FESOM | KPP | $Ri_{crit} = 0.27$ |
| I_TKE_01 | ICON-O | TKE | $c_k = 0.1$ |
| I_TKE_02 | ICON-O | TKE | $c_k = 0.2$ |
| I_TKE_03 | ICON-O | TKE | $c_k = 0.3$ |
| I_KPP_030 | ICON-O | KPP | $Ri_{crit} = 0.3$ |
| I_KPP_027 | ICON-O | KPP | $Ri_{crit} = 0.27$ |
| I_TKE_02_Langmuir | ICON-O | TKE | $c_k = 0.2$, additional Langmuir parameterisation (Axell, 2002) |
| I_TKE_02_minTKE | ICON-O | TKE | $c_k = 0.2$, minimum background TKE = $10^{-5}$ J/kg (default: $10^{-6}$ J/kg) |
| I_TKE_02_minKv | ICON-O | TKE | $c_k = 0.2$, minimum background diffusivity = $10^{-5}$ m$^2$/s (viscosity = $10^{-4}$ m$^2$/s) |
| I_TKE_02_FBF | ICON-O | TKE | $c_k = 0.2$, FESOM default bulk formulae |

## 3 Description of vertical mixing schemes

### 3.1 TKE scheme

The TKE scheme (Gaspar et al., 1990) is a commonly used vertical mixing scheme in OGCMs. The scheme that Gaspar et al. (1990) propose is based on the turbulence closure schemes of Mellor and Yamada (1974), which requires solving a prognostic turbulent kinetic energy (TKE) equation. Gaspar et al. (1990) adapted the scheme for the ocean and used a new formulation for the mixing length which had been developed by Bougeault and André (1986) and Bougeault and Lacarrere (1989).

The TKE scheme uses the classical eddy diffusivity concept to parameterise the turbulent vertical fluxes, assuming for example for temperature:

$$\overline{-T'w'} = k_v \cdot \frac{\partial \overline{T}}{\partial z} \tag{1}$$

where $T$ denotes temperature, $w$ the vertical velocity, the overbar means a time mean and the dash means deviations from this mean as obtained by Reynolds averaging. Hence, $\overline{-T'w'}$ is the turbulent vertical flux of temperature, and this is parameterised by assuming that the small scale turbulence behaves like diffusion and using the (vertical) eddy diffusivity $k_v$. The same can be done for the turbulent vertical flux of velocity **u** using the (vertical) eddy viscosity $A_v$.

The eddy viscosity and diffusivity can be obtained from the turbulent kinetic energy (TKE) as follows:

$$A_v = c_k \cdot L_{mix} \cdot E_{tke}^{1/2} \tag{2}$$

 and

$$k_v = A_v/Pr \tag{3}$$

where $c_k$ is a constant, $L_{mix}$ is the mixing length, $E_{tke}$ is the turbulent kinetic energy, and $Pr$ is the turbulent Prandtl number. It is unclear what value is best for $c_k$, and part of this study is to look at the effect of varying it. Gaspar et al. (1990) suggest a value of $c_k = 0.1$, and observational values of the mixing efficiency in the ocean which $c_k$ depends on suggest that it should not be larger than 0.3 (Deppenmeier et al., 2020). However, higher values have been tried, e.g. 0.5 by Deppenmeier et al. (2020). The turbulent Prandtl number $Pr$ is just set to 1 in Gaspar et al. (1990), but can also be set to vary with the Richardson number $Ri$. We use the CVMix default of $Pr = 6.6Ri$. The mixing length $L_{mix}$ can be thought of as the maximum length that a particle can be moved against the stratification by the turbulent motion, and thus it depends on the kinetic energy of the turbulent motion and on the stratification of the surrounding water. The turbulent kinetic energy $E_{tke}$ is determined by the prognostic TKE equation, which is integrated by the model together with the primitive equations.

The breaking of internal waves in the ocean interior is parameterised by setting a constant minimum value of TKE. The diffusivity is then still dependent on $N^2$. We use a constant minimum TKE value of $E_{tke,min} = 10^{-6}$ m$^2$ s$^{-2}$, as suggested in Gaspar et al. (1990). Additionally, we do a test run with an enhanced minimum TKE value of $E_{tke,min} = 10^{-5}$ m$^2$ s$^{-2}$, as well as a different test run with a minimum background diffusivity and viscosity, which then do not depend on $N^2$. These two runs were only done with ICON-O to save computational expenses.

Another ICON-O run was done with an additional extension of the TKE scheme: the parameterisation of Langmuir turbulence (Axell, 2002). Langmuir turbulence, which is generated through the interaction of wind-driven surface currents and wind-generated surface waves, is responsible for additional turbulent energy input into the upper ocean, and it has been shown to be important over much of the global ocean area (e.g. Belcher et al., 2012). Since we do not simulate surface waves with ICON-O, the effect of the Langmuir turbulence is missing if it is not parameterised. A limitation of the Langmuir turbulence scheme used here is that the Stokes drift is estimated from the wind stress, because there are no waves in the model.

## 3.2 KPP scheme

The nonlocal K-Profile Parameterisation (KPP, Large et al., 1994) is based on specifying vertical profiles of the eddy diffusivity and viscosity in the ocean boundary layer. As in the TKE scheme, the eddy diffusivity concept is applied, but the KPP scheme additionally includes a nonlocal term to parameterise e.g. convection. As in Gutjahr et al. (2021), it is assumed that the local and nonlocal eddy diffusivity are equal. The local eddy diffusivity is calculated as the product of a turbulent velocity scale $\omega$ and a non-dimensional vertical shape function $G$ which both depend on the normalised boundary layer depth $\sigma$:

$$k_v(\sigma) = h\omega(\sigma)G(\sigma) \tag{4}$$

where $\sigma = z/h$ is a dimensionless depth coordinate varying between 0 and 1 in the boundary layer, with $z$ denoting the depth below the surface and $h$ the ocean boundary layer depth.

The boundary layer depth $h$ is defined as the depth $z$ at which the bulk Richardson number becomes larger than a critical Richardson number $Ri_{crit}$. This is usually set to $Ri_{crit} = 0.3$. However, the critical bulk Richardson number below which

the water column becomes unstable should be dependent on vertical resolution and approach the critical gradient Richardson number of 0.25 as the vertical resolution becomes higher. Since we run the models at relatively high vertical resolution in the upper ocean here, we did two different KPP runs with $Ri_{crit} = 0.3$ and $Ri_{crit} = 0.27$, respectively, to test the effect of a reduced Richardson number threshold.

Below the boundary layer, we use the Richardson-number dependent PP scheme (Pacanowski and Philander, 1981).

## 4 Large scale tropical Atlantic structure

### 4.1 Mean tropical mixed layer depth

An important metric to evaluate the performance of the vertical mixing parameterisation is the depth of the surface ocean mixed layer. In Figure 1, the 2015 annual mean mixed layer depth (MLD) in the tropical Atlantic is shown for ICON-O and FESOM (with TKE, $c_k = 0.2$) together with a climatology derived from Argo float observations (Argo, 2022). The MLD has been calculated from the models and all available Argo float profiles in the tropical Atlantic using a density criterion. To determine MLD in both the models and the Argo float data, we use a threshold value of 0.125 kg m$^{-3}$ (e.g. Levitus, 1982) for the increase in potential density relative to the potential density at 5 m depth. It is recommended to compute MLD in OMIP and CMIP models using a threshold value of 0.03 kg m$^{-3}$ (Griffies et al., 2016; see also the discussion by Treguier et al., 2023). However, in the tropics, this threshold corresponds to a temperature difference of less than 0.01 K, which may reflect MLD changes due to diurnal warming in the near-surface layer – an effect that is poorly represented in the Argo reference data. Nevertheless, the results are comparable when using 0.03 kg m$^{-3}$ as a threshold (not shown), except that the MLD is then generally shallower in the model runs and the reference Argo dataset. As visible in Figure 1, both ICON-O and FESOM simulate too shallow MLDs in the equatorial Atlantic compared to the Argo float climatology. In Figure 2, the difference to the Argo MLD is shown for each of the model runs with the different vertical mixing settings. All sensitivity runs have a too shallow MLD for most parts of the tropical Atlantic as well. Interestingly, the bias pattern in the FESOM runs is quite similar for all runs, whereas for ICON-O, the difference between TKE and KPP is more pronounced, with the ICON-O KPP runs even showing a narrow region of too large MLD north of the equator. However, the parameter changes within the two mixing schemes hardly affect the MLD bias pattern in both FESOM and ICON-O.

The MLD bias is especially large close to the equator. Figure 3 therefore shows the mean MLD along the equator (averaged between 4°S and 4°N to include the cold-tongue region) for the different model runs together to provide a better quantitative comparison. As seen from Figure 2, KPP is much closer to observations than TKE for ICON-O on the equator, whereas the difference between TKE and KPP is negligible in FESOM. In ICON-O with smaller $c_k$, the equatorial MLD becomes slightly deeper, i.e. more realistic, although not nearly to an extent that could remove the bias. This behavior is at first counterintuitive, because a larger $c_k$ should lead to larger viscosity and diffusivity (Eq. 2). However, the change in $c_k$ of course also leads to differences in the density and current structure, which changes the amount of TKE and can thus eventually lead to nonlinear changes in mixing. The main factor leading to a larger equatorial MLD with smaller $c_k$ is most likely the Equatorial Undercurrent, which is a source of shear instability but weakens with increasing $c_k$. We explore this process in more detail in Section 4.3.

Changing the critical Richardson number in the KPP scheme has almost no effect in both models. The three additional ICON-O runs with enhanced background turbulence and Langmuir turbulence have a very similar equatorial MLD to the I_TKE_02 run (not shown).

## 4.2 Temperature

### 240 4.2.1 Mean surface temperature distribution

As described in the introduction, the tropical SST affects atmospheric convection, and thus can influence large-scale tropical and also extratropical wind and precipitation patterns. It is thus quite important to simulate the tropical SST distribution well in climate models. In Figure 4, the annual mean 2015 sea surface temperature in the tropical Atlantic is shown for HadISST on the left, and the difference to HadISST in the different model runs in the centre column (ICON-O) and right column (FESOM).
In both models, a typical warm bias is evident in the upwelling regions along the African coast, which has long existed in most ocean and climate models (e.g. Farneti et al., 2022). Compared to CMIP6 models, the warm bias in the eastern upwelling regions and in the eastern equatorial region in ICON-O and FESOM is rather small, maybe owing to the high horizontal resolution used in this study – Richter and Tokinaga (2020) and Farneti et al. (2022) find that the warm SST biases in the eastern tropical Atlantic in HighResMIP are reduced compared to standard CMIP6. As shown by Farneti et al. (2022), the SST
bias at the eastern equatorial Atlantic coast is about 2 to 3°C in the CMIP6 multi-model mean and about 2°C in the HighResMIP multi-model mean. Compared to this, the eastern equatorial Atlantic SST biases are even smaller in our ICON-O and FESOM runs, with about 0.5°C in ICON-O and FESOM. However, the largest bias in ICON-O and FESOM is not the warm bias in the east, but a very strong cold bias in the central and western tropical Atlantic. This cold bias has a similar pattern in both models, with an intensification on the equator, but is considerably stronger in ICON-O (-1.5 to -2°C) than in FESOM (only about -1°). 
The cold SST bias in the western equatorial Atlantic is stronger in our ICON-O and FESOM runs than typically seen in GCMs, with -1 to -2°C for FESOM and ICON-O, compared to about 0 in the CMIP6 and HighResMIP multi-model means (Farneti et al., 2022).

Changing the vertical mixing scheme or associated parameters has almost no effect on the mean SST bias in the tropical Atlantic, especially in FESOM. In ICON-O, changes between some of the different runs are visible, but with no effect on the
large-scale bias pattern. In all cases, the western equatorial cold bias remains larger in ICON-O than in FESOM. The eastern warm bias is reduced in ICON-O when using KPP. In the west, the cold bias becomes larger in the ICON-O TKE runs when using a large value of $c_k = 0.3$.

### 4.2.2 Seasonal cycle of SST in the tropical Atlantic

The SST in the tropical Atlantic shows strong seasonal variations, especially in the eastern equatorial Atlantic. There, the
Atlantic Cold Tongue (ACT) develops every year in boreal summer, when the trade winds intensify due to the northward movement of the Intertropical Convergence Zone (ITCZ) and the shear between surface and subsurface currents increases and leads to vigorous turbulent mixing that cools the surface (Hummels et al., 2014). Interannual variations of this seasonal

cycle are related to the Bjerknes feedback and are generally referred to as Atlantic Niño or Atlantic Zonal Mode. The ACT and Atlantic Niño are of large importance for the surface climate of the surrounding continents, but GCMs generally have
difficulties with reproducing them (e.g. Lübbecke et al., 2018).

In Figure 5, the seasonal cycle of SST averaged over the ATL3 box in the eastern tropical Atlantic is shown, for HadISST in black and the different model runs in colour. The development of the ACT is visible in HadISST as a strong decline in temperature starting in April/May, with the ATL3 SST reaching a minimum in August. Both models follow this seasonal cycle with a somewhat too fast temperature decline, so that the minimum is already reached in July. The vertical mixing scheme and
its parameter settings do not have a large effect on the seasonal cycle of ATL3 SST in either model. The largest differences can be seen in the minimum temperature in July and August, when the cold tongue is strongest. The KPP runs are closest to observations, while the TKE runs are slightly colder; the coldest runs are those with the smallest $c_k$ in both models.

In the western tropical Atlantic, the situation is different (Figure 6). It is noticeable that the models generally reproduce the seasonal cycle relatively well, but with a systematic offset towards colder temperatures compared to HadISST. In FESOM, the
SST difference to HadISST is up to 1°C, in ICON-O about twice as large. The large difference between ICON-O and FESOM might partly be due to the different sets of bulk formulae used, which is investigated in Section 6. Interestingly, the SST bias in the western Atlantic is sensitive to the vertical mixing scheme configuration in ICON-O, whereas in FESOM all the different runs are very close to each other as it was for both models in the eastern tropical Atlantic (Figure 5). In ICON-O, the amplitude of the seasonal cycle is overestimated more when using KPP than TKE. The cold bias of the model SST is especially strong
when using TKE with a high value of $c_k = 0.3$, and about 0.5°C less when using $c_k = 0.1$ instead. However, even though distinct changes between the different ICON runs are visible, they are still smaller than the differences between the models and between models and HadISST.

The seasonal cycle of ATL3 and WATL SST in the additional runs with ICON-O (with the Langmuir parameterisation and an enhanced background TKE/diffusivity) looks very similar to I_TKE_02 (not shown).
The seasonal variation in the monthly mean SST in the equatorial Atlantic (averaged between 3°S and 3°N to facilitate comparison with a similar analysis of OMIP simulations by Prigent and Farneti, 2024) is displayed as a series of Hovmoeller plots in Figure 7. Comparing the 25°C contours between the models and HadISST, it is clear that the cold tongue extends too far westward from July to October in ICON-O and, to a lesser extent, in FESOM. Similarly, comparison of the 28°C contour shows that the SST in the western equatorial Atlantic is also too low in both ICON-O and FESOM between March to July. In contrast
to the 25°C contour, the westward extent of the 28°C contour is fairly similar for both ICON-O and FESOM, suggesting that model differences in the annual mean SST bias (Figure 4) are due to excessive equatorial cooling in the summer months. The equatorial SST in ICON-O and FESOM simulations stand in contrast to the ocean-only OMIP1 and OMIP2 simulations, where the summer cooling is underestimated and the spring warming is overestimated (Prigent and Farneti, 2024).

The corresponding biases for the monthly mean SST relative to the HadISST data are shown in Figure 8. The erroneously
large westward extent of the cold tongue is evident from the strong cold biases west of 10°W in both ICON-O and FESOM. The cold biases are most pronounced around June and July, the timing of which is consistent with the cooling in ICON-O and FESOM occurring too fast compared to observations as seen in Figure 6 for the ATL3 box. The larger cold biases in ICON-O

compared to FESOM are also consistent with a perennially lower SST in ICON within the WATL box than in FESOM (see 6).
Warm biases appear in the eastern half of the ATL3 box between 10°W and 0°W and peak around September, which points

to an early onset of warming also visible in Figure 5 in this period. Interestingly, the warm bias maximum tends to occur
before the cold bias maximum in coupled GCMs - the reverse of what we see in ICON-O and FESOM (Richter and Tokinaga,
2020). As shown in Figure 4, the cold biases display more variation between simulations for ICON-O than for FESOM, with
the ICON-O TKE simulation using $c_k = 0.3$ showing the strongest cold bias. Interestingly, increasing $c_k$ in ICON-O not only
increases the magnitude of the JJA cold bias west of 10°W but also the September warm bias between 10°W and 0°W. For

FESOM on the other hand, increasing TKE decreases the magnitude of the JJA cold bias but increases the magnitude of the
warm September bias. As seen before, the changes due to the mixing parameter changes are small compared to the magnitude
of the bias.

### 4.2.3   Mean vertical temperature distribution

In Figures 9 and 10, vertical sections of the annual mean 2015 temperature are shown for the upper 200 m of the tropical

Atlantic. The panel on the left shows a mean temperature section from Argo float data, the centre panels show the difference
to Argo for ICON-O, the right panels for FESOM.

Along the equator, there is a cold bias above the thermocline and a warm bias below it in all model runs. Farneti et al.
(2022) also show the subsurface temperature bias along the equator for OMIP2, CMIP6 and HighResMIP (their Figure 6).
The equatorial subsurface temperature bias in our ICON-O and FESOM runs are rather atypical compared to the OMIP2 multi

model mean bias, which is too warm in the upper 200 m. The bias in ICON-O and FESOM rather resemble the multi-model
mean subsurface temperature bias from CMIP6 and HighResMIP. For CMIP6 and HighResMIP, the multi-model mean bias
shows a too cold wedge between the surface and about 100-150 m depth extending from the west to the central-eastern tropical
Atlantic. Below this and in the east, there is a warm bias. The cold western subsurface equatorial temperature bias in CMIP6
is about -2°C, in HighResMIP about -3°C, in our ICON-O runs between -2 and -4°C, and in our FESOM runs between -1

and -2°C. Unlike in the HighResMIP and CMIP6 model means, in ICON-O and FESOM the subsurface cold bias extends all
the way to the east, which is rather atypical. The warm equatorial subsurface temperature bias in the east is about 2°C in the
CMIP6 multi-model mean, about 1°C in HighResMIP, about 2 to 4°C in our ICON-O runs, and about 1 to 2°C in our FESOM
runs (the warm eastern subsurface bias is also shifted and intensified to the west in ICON-O and FESOM compared to the
CMIP6 and HighResMIP multi model means, Farneti et al., 2022).

The biases are again generally stronger in ICON-O than in FESOM, as for the mean SST. For the warm biases, the maximum
values shown in Figures 9 and 10 for the ICON-O runs are 40 to 140% higher than for the equivalent FESOM runs. For the
cold biases, the minimum values in the ICON-O range from 10 to 60% lower than for FESOM. In the warm bias below the
thermocline, some reactions to changes in the mixing scheme can be seen that are consistent between the two models: the
bias becomes stronger when $c_k$ is increased in the TKE scheme, and it is also stronger in KPP than in TKE (although this is

not so obvious in FESOM). As seen before, the changes between the different FESOM runs are actually not very large, while
ICON-O reacts more sensitively to changes in the vertical mixing parameterisation. Including the Langmuir parameterisation

does not have a large effect. Increasing the background TKE has a similar effect as increasing $c_k$ in the TKE scheme: both the cold bias above the thermocline and the warm bias below the thermocline increase (i.e. the thermocline becomes more diffuse). Increasing the minimum background diffusivity does not increase the bias as much.

A near-surface cold bias with a warm bias below is also visible along 23°W, which coincides with the PIRATA mooring. However, the biases are generally weaker in the northern hemisphere and show more variability with latitude than with longitude. This is in accordance with the much larger zonal than meridional scales of the tropical Atlantic circulation systems. Again, there are local differences between the different mixing scheme settings, but not enough to change the large scale bias pattern. As in the sections along the equator, the difference between KPP and TKE is larger in ICON-O than in FESOM.

## 4.3 Equatorial current system

The equatorial oceans are characterized by strong zonal current systems. An important part of these is the Equatorial Undercurrent (EUC), which is one of the strongest subsurface currents in the world oceans. It transports water eastward approximately at the depth of the thermocline along the equator, and contributes to the Meridional Overturning Circulation (MOC) and the Subtropical Cells (e.g. Johns et al., 2014; Brandt et al., 2021). The EUC is not only important for horizontal water mass trans-

port, but also for vertical mixing in the equatorial ocean: due to the shear between the mean westward surface flow of the South Equatorial Current and the eastward flow of the EUC at thermocline depth, the upper equatorial water column is permanently close to a state of marginal instability. This has some interesting consequences for the generation of turbulence in the equatorial oceans, which is diurnally enhanced not only in, but also below the surface mixed layer (Smyth et al., 2013; Moum et al., 2022a). On the other hand, it has been shown that the EUC is notoriously hard to simulate in ocean models (e.g. Karnauskas

et al., 2020; Fu et al., 2022), and that it reacts relatively sensitively to e.g. the vertical mixing parameterisation (e.g. McCreary, 1981). We therefore take a closer look at how the Atlantic EUC is represented in ICON-O and FESOM and how it reacts to the different vertical mixing scheme settings.

   In Figure 11, cross sections of the mean zonal velocity along 23°W are shown. The first panel shows a multi-year mean from shipboard observations during 21 cruises (available at https://doi.pangaea.de/10.1594/PANGAEA.899052, Burmeister et al.,

2019). The Equatorial Undercurrent (EUC), a very strong eastward subsurface current, can be clearly seen on the equator with its core at approximately 70 m depth. In the cruise observations, it has a mean core velocity of $0.79\,\mathrm{m\,s^{-1}}$. The panels in the centre column show the same for the different ICON-O runs, the panels on the right for FESOM.

   In general, the EUC is too weak as well as too deep and broad in ICON-O, whereas it is closer to the observed strength and location in FESOM. This indicates that the horizontal friction in ICON-O might be too large, making the EUC generally too

weak even with the same vertical friction settings as in FESOM. In both models, increasing $c_k$ in the TKE scheme leads to a weakening and deepening of the EUC. The weakening is likely related to an increased eddy viscosity with increased $c_k$ (not shown). Since the EUC core depth is strongly related to the depth of the thermocline, the deepening of the EUC core is likely related to changes in the stratification with increasing $c_k$. This is consistent with the mean temperature bias sections along the equator shown in Figure 9 where a strong influence of $c_k$ on the vertical temperature gradient in the EUC depth range can be

seen. In ICON-O, the EUC is most realistic with the smallest tested value of $c_k = 0.1$, although it is then still a bit too weak

and too deep. Since the EUC is in general stronger and shallower in FESOM, it even becomes too strong with $c_k = 0.1$, so that the most realistic EUC for FESOM is obtained with $c_k = 0.2$.

Changing the vertical mixing scheme from TKE to KPP has a very different effect in ICON-O and FESOM. In ICON-O, the EUC strength reduces dramatically to below $0.3\,\mathrm{m\,s^{-1}}$. With a lower critical bulk Richardson number of $Ri_{crit} = 0.27$, the reduction is even stronger. In FESOM, the EUC core velocity with KPP remains in the range of values for the different TKE runs, and with $0.71\,\mathrm{m\,s^{-1}}$ both for $Ri_{crit} = 0.3$ and $Ri_{crit} = 0.27$ reasonably close to the observed value of $0.79\,\mathrm{m\,s^{-1}}$.

Adding the Langmuir turbulence parameterisation to the ICON-O TKE run makes the EUC slightly stronger, as does enhancing the background diffusivity. Increasing the minimum background TKE instead makes the EUC slightly weaker.

ICON-O and FESOM do a reasonable job of capturing the seasonality of the EUC. Figure 12 shows the the monthly-mean zonal velocity at 23°W as a function of depth averaged between 3°S and 3°N. The equatorial zonal velocity in the models displays two peaks, one in MAM and one in SON. The timing of these two peaks is in good agreement with observations at 23°W (Johns et al., 2014). Moreover, both models qualitatively capture the observed deepening of the EUC in the summer months. The EUC deepens by about 30 to 50 m in the models. The main difference between ICON-O and FESOM is the relative strength of the two peaks in zonal velocity. In ICON-O, the SON peak is stronger by about 10 cm/s and prolonged compared to the MAM peak. In FESOM by contrast, the two peaks are of similar magnitude and duration. Similar to the longitudinal cross section of annual-mean zonal velocity shown in Figure 11, ICON-O displays larger intra-model differences than FESOM. Note that while the maximum zonal velocity increases with increasing $c_k$ for the ICON-O TKE simulations in Figure 11, the reverse appears to be true for Hovmoeller plot in Figure 12. This discrepancy is presumably due to the increasing latitudinal extent of the EUC with increasing $c_k$.

## 5 Representation of small scale processes in the upper ocean

One significant source of energy for the ocean system is wind energy. The way this wind energy is transferred from the atmosphere to the ocean depends on the physical setting in the upper ocean layers, e.g. stratification, which is both set by and impacts mixing. In the following, we will take a closer look at some of the processes distributing wind energy vertically in the ocean. In particular, the energy distribution within the mixed layer is crucial for modeling the energy and water cycle of the Earth's entire climate system.

### 5.1 Near Inertial Waves

Near-inertial waves (NIWs) are significant contributors to wind-driven diapycnal mixing in the ocean (e.g. D'Asaro, 1985; Alford, 2003; Zhai et al., 2009). Estimates of wind work done on near-inertial motions in the global ocean range from 0.3 to 1.6 TW (Alford et al., 2016). These waves are confined to the mixed layer and oscillate horizontally with period corresponding to the local inertial frequency and speeds of up to 1 ms$^{-1}$. At 15N, the period of NIWs is of about two days and increases toward the equator. Resonant tropical cyclonic winds induce strong near-inertial currents. Therefore, NIWs play a significant

role in vertical mixing in the tropical northern Atlantic, especially during boreal autumn when the mixed layer is thinnest, and the wind speeds are highest (Foltz et al., 2020).

In Hummels et al. (2020), mixing induced by a NIW in the North Atlantic Ocean is reported and analysed. The wave cools the mixed layer at the rate of $244\,\mathrm{Wm^{-2}}$, deepens it and induces mixing below it. The observations of this extreme mixing event provide a unique challenge for the mixing schemes. A more extensive analysis of the NIW structure is available in Mrozowska et al. (2024). Both ICON and FESOM fail to reproduce the observed NIW amplitude. Here, we present the models' ability to simulate the stratification changes observed as a result of NIW-induced mixing.

The site of the observations from Hummels et al. (2020) is located at $11°$N, $21°$W. The 25 microstructure profiles were collected over the course of 24 hours between the 13th and 14th of September, 2015, during the R/V Meteor cruise M119 (Fischer, 2020). The vertical resolution of the measurements is approximately 0.5m.

Snapshots of the buoyancy frequency ($N^2$) in the observations and the models are presented in Figure 13. The depth of the mixed layer in the TKE runs at the site is generally shallower than in the KPP runs. Both I_KPP and I_KPP_027 reproduce the observed $N^2$ profiles within the mixed layer most accurately. The base of the mixed layer is highly stratified in the TKE models, with $N^2$ values reaching three times the observed, and with diffusivity values as low as $10^{-7}\mathrm{m^2s^{-1}}$. The only exceptions are the I_minTKE and I_minkv results, where minimum background TKE and diffusivity are imposed, respectively. The TKE tuning does not affect the depth of the mixed layer at the site. Significant differences between ICON-O and FESOM are noticeable: the high stratification band is weaker in FESOM TKE, and ICON-O KPP simulates a mixed layer which is twice as deep as in FESOM KPP.

The simulated $N^2$ profile in this extreme NIW-induced mixing event is most sensitive to the mixing scheme chosen. KPP reproduces a more realistic stratification within the mixed layer. The vertical structure of the $N^2$ profiles is not markedly affected by the tuning of TKE parameters.

## 5.2 Diurnal cycle of upper ocean properties

The diurnal cycle is the most dominant variability on timescales shorter than the inertial rotation of the upper ocean. It is particularly pronounced in the tropics, where a large amount of high shortwave solar radiation penetrates the upper ocean during the day, heating it up. Under low wind conditions, a stable stratification known as the diurnal warm layer (DWL) often forms in the surface layer. The depth of this DWL ranges from several centimeters to tens of meters, depending on factors such as wind conditions and incoming solar radiation. The presence of a DWL directly affects how momentum from the wind is distributed in the upper ocean. In the following, we separate between off-equatorial regions and the equator itself for two reasons. First, the corresponding processes depend, to some extent, on the Coriolis force. Second, the equatorial region experiences considerable velocity shear due to the EUC (section 4.3), which interacts with the processes occurring on daily timescales.

### 5.2.1 Off-equatorial regions

Off the equator, the presence of a DWL directly affects how momentum from the wind is distributed in the upper ocean. When a DWL is present (resulting in strong stratification just below the ocean surface), an ocean current forms downwind, known as a diurnal jet. A significant portion of wind energy therefore remains in the uppermost layers of the ocean. In the absence of a DWL (resulting in weak stratification), no diurnal jet forms and the momentum is distributed over the entire extent of the mixed layer (ML). This diurnal jet of surface water influences wind stress and wind power input, thereby affecting the exchange of properties such as momentum, moisture, and heat between the ocean and the atmosphere. Air-sea fluxes are partially dependent on the surface water velocity aligned with the wind direction, which itself is influenced by the stratification caused by the diurnal warm layer (DWL). When the surface flow deviates from the wind direction due to Coriolis deflection, the impact on air-sea fluxes diminishes. In the (sub-)tropics, the surface flow is usually aligned with the wind direction, making DWLs an important component of the energy/heat and freshwater input to the ocean. A high vertical resolution within the ML (at least 3-4 depth levels) is required to model these processes at all. Fig. 14 shows composites of daily temperature anomalies of the upper-ocean modelled with the different ICON-O and FESOM runs compared to temperature observations from three Slocum gliders. All data are averaged over January and February and shown for the subtropics in the western tropical Atlantic (between 12°N-14.5°N and 56.5°W-59°W). First, the diurnal temperature variations in both models align remarkably well with observations. However, subtle differences do emerge. It is noteworthy that the difference between the models is greater than that among the different mixing parameterisations. The ICON-O model represents the daily temperature cycle in the upper ocean more realistically than FESOM. In the FESOM model, the diurnal temperature variation is slightly too small (compare especially the center-top plot in Fig. 14, where the reddish lines are closer to the observations (black) than the bluish lines) and the DWL does not extend as deeply. When comparing the differences between the various runs, the diurnal temperature cycle becomes more pronounced and deeper in FESOM with increasing $c_k$. KPP behaves similarly to TKE with $c_k = 0.2$, but appears to mix the diurnal warm layer (DWL) more rapidly in the evening. This quicker mixing is more realistic compared to the observations. Among the ICON-O runs there is no distinct pattern for variations in $c_k$. However, KPP generates too little diurnal temperature variation. Over all, the ICON-O with $c_k = 0.3$ represents the observed diurnal temperature cycle in the upper ocean best.

### 5.2.2 Deep cycle turbulence on the equator (0°N, 23°W)

A unique aspect of equatorial turbulence is the existence of diurnally-varying turbulence that extends well into the stratified ocean beneath the surface mixed layer (Gregg et al., 1985; Moum and Caldwell, 1985). This sub-mixed layer turbulence has been termed deep cycle (DC) turbulence. DC turbulence is present in the equatorial Atlantic as well as in the Pacific extending from the base of the mixed layer to the core of the EUC while exhibiting elevated turbulence during night time (e.g. Moum et al., 2022a). Its magnitude is governed by surface wind stress, local shear between the mixed layer and the EUC core and solar buoyancy flux (Smyth et al., 2021; Moum et al., 2023). It has been shown that global ocean general circulation models are capable of simulating DC turbulence (e.g. Pei et al., 2020).

Here, the model output is compared to time series of turbulence recorded by $\chi$-pods that were attached to a PIRATA buoy at 0°N, 23°W in the Atlantic and are available for the period of the model runs (Moum et al., 2022b). We focus on the ability of the models to simulate enhanced mixing penetrating below the mixed layer (Fig. 15), its timing and descending rate (Fig. 16), which in the observations is close to 6 m/hour consistent with laboratory studies of entrainment in stratified shear flows (Moum et al., 2023), and the magnitude of eddy diffusivities below the mixed layer (Fig. 17).

None of the different model runs represent all aspects of DC turbulence. While the ICON TKE runs exhibit descent rates and maximum penetration depths comparable to that of the observations (upper three left panels in Fig. 15, 16) they fail to reproduce its daily rhythm. Furthermore, average maximum eddy diffusivities in the thermocline in the ICON TKE runs are about a factor of 10 larger than suggested by the observations (Fig. 17). In contrary, the FESOM TKE runs accurately reproduce the average maximum thermocline diffusivities (upper three right panels in Fig. 17) but fail to show downward propagation of night time turbulence (Fig. 16). Finally, the ICON and the FESOM KPP runs reproduce the observed eddy diffusivity maximum and its depth distribution in the thermocline quite well (bottom two left and right panels in Fig. 17) but fail to show downward propagation of their strongly elevated mixed layer diffusivities (Fig. 15). However, the timing of maximum eddy diffusivity in the thermocline somewhat agrees with the observations (Fig. 16).

In summary, TKE in ICON can capture the actual propagation process suggesting that TKE is able to model the physical processes involved in DC turbulence more realistially. In the spirit of the nextGEMS philosophy of gaining insights beyond tuning, this would be a strong argument in favour of using TKE in ICON. However, for integral performance aspects, such as vertical heat and salt transport, the DC turbulence propagation itself is likely irrelevant. Instead, a realistic representation of the vertical distribution of the maximum of eddy diffusivity is important, as found for KPP, but also when using TKE in FESOM. However, this comes with the uncomfortable thought that the correct result may have been achieved for the wrong reasons.

## 6 Importance of forcing bulk formulae for model differences

As shown in the sections above, we generally find larger differences between the two models that we used than between different settings of the vertical mixing scheme. We were therefore interested in where the large differences between FESOM and ICON-O come from. Because of limited resources, we could test only one possible factor, and we chose the one that to us seemed to have most potential for a big impact on the mean upper ocean state: the different sets of bulk formulae in ICON-O and FESOM.

To quantify the effect of the different bulk formulae, we did an additional test run with ICON, with the same settings as in I_TKE_02, but with the bulk formulae that FESOM uses (for details see Section A). We call this run I_TKE_02_FBF. Note that this is only done as a sensitivity test to investigate how large the influence of differences in the bulk formulae can potentially be compared to the changes that we tested in the mixing scheme, not to get a more realistic result. Both models have been tuned to work well with their respective sets of bulk formulae and would therefore have to be retuned to work well with a different set. The effect that using the FESOM bulk formulae in ICON-O has on the mean tropical Atlantic sea surface temperature, mixed layer depth, and Equatorial Undercurrent strength, can be seen in Figures 18, 19 and 20.

The sea surface temperature and the Equatorial Undercurrent strength react to the different bulk formulae. The SST pattern in ICON-O becomes more similar to FESOM when using the same bulk formulae, but interestingly, the entire tropical Atlantic also becomes much warmer. While ICON-O with its default bulk formulae has a much larger cold bias in the western Atlantic than FESOM, its western cold bias reduces very much and is smaller than FESOM's when using the FESOM bulk formulae. Instead, the eastern Atlantic warm bias is then much stronger than before and also much stronger than in FESOM. For the EUC, the difference between the models reduces when using the same set of bulk formulae, i.e. the EUC in ICON-O gets stronger. However, it is still significantly weaker than in FESOM and observations.

The mixed layer in the northwest of the equator becomes deeper in ICON-O when using the FESOM bulk formulae, getting closer to the FESOM MLD pattern. However, the equatorial MLD bias stays much larger than in FESOM even with the changed bulk formulae. Here, the bulk formulae do not seem to make much difference, suggesting that other factors lead to the different mean MLD pattern in the model. These could include differences in lateral mixing, in the horizontal grid, or in the numerical schemes and the associated numerical mixing.

The different bulk formulae that are by default applied in FESOM and ICON-O can explain a significant part of the models' differences in mean tropical Atlantic SST and EUC strength, while they do not affect the mean mixed layer depth much. However, also for SST and EUC, a large unexplained difference remains such that other factors must contribute as well.

## 7 Discussion

Earlier studies, e.g. by Blanke and Delecluse (1993); Deppenmeier et al. (2020) and Zhang et al. (2022), have suggested that the long-standing tropical Atlantic ocean model biases can be significantly reduced by changes in the vertical mixing scheme. However, these studies focused on a specific model and usually only on a specific process or variable. Gutjahr et al. (2021) made a more comprehensive comparison of the effect of four different mixing schemes, albeit also only in one model. They found that the overall effect of the mixing scheme, e.g. on the global surface temperature bias, was rather small and at most relevant for very limited regions.

Although we can partly reproduce some of the findings of Blanke and Delecluse (1993), Deppenmeier et al. (2020) and Zhang et al. (2022), we show that overall, the large scale patterns and the magnitude of the tropical Atlantic mean state biases are not sensitive to the changes in the vertical mixing scheme settings that we tested. This result is largely consistent with the conclusions of Gutjahr et al. (2021). In the cases where the biases are sensitive to the vertical mixing scheme settings, they are model dependent. The Equatorial Undercurrent (EUC), for example, is much less well reproduced in ICON-O when using KPP compared to TKE. Because we use the KPP scheme in the surface boundary layer together with the Richardson number dependent PP scheme in the deeper ocean, this is consistent with the findings of Blanke and Delecluse (1993) who found that the EUC is better reproduced when using TKE compared to PP. In contrast, the EUC is realistically simulated in FESOM with both KPP and TKE. Deppenmeier et al. (2020) claim that they can decrease the surface temperature bias in the tropical Atlantic in the NEMO ocean model by increasing $c_k$ in the TKE scheme. In their study, increasing $c_k$ from 0.1 to 0.5 leads to a surface cooling in the eastern tropical Atlantic and a subsurface warming in the tropical Atlantic. In ICON-O, the SST also becomes

colder when increasing $c_k$, but in the western instead of the eastern equatorial Atlantic, reinforcing the western cold bias of the model. The subsurface warming also happens in ICON-O and FESOM when increasing $c_k$, and again, in these two models this actually leads to a bias increase because it reinforces the original bias pattern. We can thus confirm that some of the sensitivity

of the tropical Atlantic surface and subsurface temperature to the value of $c_k$ as described by Deppenmeier et al. (2020) is present also in ICON-O and FESOM, but whether it is beneficial to increase or decrease $c_k$ is heavily dependent on the model, and it is also not enough to induce any significant change in the temperature bias pattern in our case. It should be mentioned that Deppenmeier et al. (2020) ran their sensitivity runs for much longer than the two years of our ICON-O and FESOM sensitivity runs. However, the magnitude of SST and subsurface temperature changes due to the $c_k$ increase are comparable in

their study and in our ICON-O runs, though much smaller in FESOM (up to about 0.5°S on the surface and up to about 1.5°C in the subsurface in their case, for FESOM much less, for ICON-O up to about 0.5°C on the surface and up to about 2°C in the subsurface). Zhang et al. (2022) focus on the tropical Atlantic subsurface warm bias, which is very similar in most Ocean Model Intercomparison Project models as in ICON-O and FESOM. They show that this warm bias can be reduced in POP2 significantly by about 2.5°C by constraining the background diffusivity in the KPP scheme to observations, i.e. reducing it by

one order of magnitude. Although we did not test this with the KPP scheme, we did similar runs with ICON-O using the TKE scheme; with the I_TKE_minKv run corresponding to the default KPP background diffusivity, and the I_TKE_02 run having up to one order of magnitude smaller diffusivities in the most parts of the interior ocean (although not in all places). We see a similar effect as Zhang et al. (2022) describe: the subsurface warm bias is increased slightly in the ICON-O TKE run with larger background diffusivity. However, the change in the subsurface temperature bias (up to about 0.2°C) is small compared

to the overall bias strength in our case, and does not change the bias pattern.

Concerning the mixed layer depth (MLD) in the tropical Atlantic, we find that in both FESOM and ICON-O, it is generally too shallow compared to Argo float observations. This is contrary to for example Zhu et al. (2022) who, in order to reduce the overly deep penetration of boundary layer mixing in MOM5, have to reduce the strength of wind stirring in their parameter settings. For the mean mixed layer depth, we again find that ICON-O is more sensitive to a change in the vertical mixing

scheme from TKE to KPP than FESOM. For ICON-O, the depth of the mixed layer near the equator becomes deeper with KPP compared to TKE, and is much closer to observations. Also in FESOM, the equatorial MLD is more realistic with KPP than with TKE, but here the effect is much smaller. Away from the equator, the MLD bias pattern is less sensitive to the vertical mixing scheme in both models.

Although it has been suggested that Langmuir turbulence is an important process for vertical mixing over much of the global

ocean (Belcher et al., 2012) and that it can improve models' representation of upper ocean mixing and mixed layer depth (e.g. Li et al., 2019), we find that including the Langmuir turbulence parameterisation from Axell (2002) in the TKE scheme in ICON-O does not really affect any of the tropical Atlantic mean state variables or small scale variability that we looked at. We especially expected it to benefit the simulation of the tropical Atlantic mixed layer depth which is too shallow in ICON-O, but the changes induced by the Langmuir turbulence scheme were in general very small or not detectable, for the MLD as

well as other variables. For the equatorial MLD, for example, the change in MLD through including the Langmuir turbulence parameterisation in ICON-O is below 1 m for most locations (more than an order of magnitude less than the mean bias). The

differences between the I_TKE_02 run and the I_Langmuir run, apart from being small, also change sign on small spatial scales not much larger than the grid scale, showing no larger-scale consistent effect on the mean biases. However, the effect of including the Langmuir turbulence parameterisation might be larger in the extratropics as found by e.g. Li et al. (2019).

For some specific processes and/or variables, the choice of the vertical mixing scheme can matter. As described above, the equatorial MLD is better in ICON-O with KPP. On the other hand, we find for example that the TKE scheme is much better suited for the simulation of the Atlantic EUC in ICON-O (though not in FESOM), and for the simulation of the deep diurnal cycle of equatorial turbulence. However, this clearly depends on the model and the process of interest, and the best vertical mixing scheme choice might be a different one when focusing on a different region or process. Changing tuning parameters in 575 either the TKE or the KPP scheme has very little effect on the tropical Atlantic mean state biases in both FESOM and ICON-O. In those cases where an effect is there, it is very small compared to the magnitude of the bias, and cannot change the large scale bias pattern.

    Because a large part of the tropical Atlantic biases are not sensitive to the vertical mixing scheme settings that we have tested (with the small exception of the aforementioned cases), the biases must instead be much more dependent on other specifics of 580 the model that is being used. This is in line with the findings of Gutjahr et al. (2021) that the choice of vertical mixing scheme can matter regionally, but that the large scale bias patterns are mainly set by other factors. One other factor that we have investigated is related to the atmospheric forcing that is used to drive the ocean model. FESOM and ICON-O use two different sets of default bulk formulae to convert ERA5 forcing fields to atmosphere-ocean fluxes. Exchanging the bulk formulae in ICON-O with those used in FESOM yielded much larger changes in the sea surface temperature field than the changes in the 585 vertical mixing scheme. This is consistent with e.g. Zhu et al. (2022) and Zhang et al. (2022) who also found that the wind forcing has a large influence on tropical temperature biases. Unlike SST, the depth of the mixed layer is not influenced by the choice of bulk formulae. However, the bulk formulae can partly explain the difference in EUC strength between FESOM and ICON-O. The fact that the atmospheric forcing plays an important role in creating the ocean model biases is not only true for uncoupled OGCMs. Also for coupled climate models, it has been shown that biases in the atmospheric model components are 590 an important source of the tropical ocean biases (e.g. Richter and Xie, 2008; Wahl et al., 2011; Richter et al., 2012; Voldoire et al., 2019).

    There are other differences in the model physics between FESOM and ICON-O beside the bulk formulae, that likely contribute to the large differences seen between the two models in all analysed variables. Although we homogenised the model settings between FESOM and ICON-O that are directly connected to the representation of vertical mixing, we left the rest of 595 the model settings as they are commonly used to see how much of the variation between the different vertical mixing settings is model specific and how much happens similarly in both models. These differences in model physics include the horizontal grid, but also lateral mixing and numerical mixing due to the numerical schemes used in the two ocean models.

    One aspect that we did not investigate here is the effect of horizontal resolution on the biases seen in the ICON-O and FESOM experiments. While the horizontal resolution of these simulations is sufficient to resolve mesoscale eddies in the 600 tropical Atlantic, the 10-km horizontal resolution is insufficient to resolve the full spectrum of sub-mesoscale features that impact vertical mixing, such as filaments. Nevertheless, the ability of the ICON-O model at 10-km resolution to capture the

vertical mixing along the sharp SST-gradients at the edges of tropical instability waves (Specht et al., 2024), suggests that sub-mesoscale mixing and the associated spatio-temporal variability are at least partially represented in the experiments.

## 8    Conclusions

We presented coordinated sensitivity experiments with two different eddy-rich ocean models, FESOM and ICON-O, to investigate the effect of changing the vertical mixing scheme settings on biases in the tropical Atlantic. The tropical Atlantic in ocean models has long been subject to large biases in the mean state, such as sea surface temperature, which limits the ability of coupled climate models to simulate tropical and extratropical climate (e.g. Richter, 2015; Lübbecke et al., 2018; Richter and Tokinaga, 2020). Previous studies attributed an important role to the parameterisation of vertical mixing in the development of

model biases of the mean state of the tropical Atlantic. Tuning of the vertical mixing scheme or a change to a different scheme thus led to a significant improvement in simulations. (e.g. Blanke and Delecluse, 1993; Deppenmeier et al., 2020; Zhang et al., 2022). However, these studies focused on a specific model and usually only on a specific process or variable.

By changing the vertical mixing schemes in two ocean models, we find that most of the long-standing biases in the large scale mean state in the tropical Atlantic Ocean are largely insensitive to the choice and details of the vertical mixing parameterisation.

For SST, subsurface temperature, and the off-equatorial MLD, we find that the bias pattern is not affected by changing the vertical mixing scheme settings, and that there is little effect on the bias magnitude. For the MLD close to the equator as well as the EUC, the mixing scheme matters more: ICON-O has a more realistic equatorial MLD with KPP, but a more realistic EUC with TKE. These sensitivities, however, are model-dependent. In FESOM, both equatorial MLD and the EUC are not changed much by switching between TKE and KPP. For smaller scale variability like the upper ocean diurnal cycle or the

representation of near-inertial waves, the choice of the vertical mixing scheme can matter as well, but the tuning of parameters within the schemes does not lead to significant improvements in diurnal cycle or near-inertial wave representation.

Our results suggest that the origin of the biases in the near-surface tropical Atlantic is complex and cannot be controlled by the ocean mixing parameterisation alone, but is likely related to biases in atmosphere-ocean interactions or the atmospheric forcing. In light of the numerous studies that have been conducted on this topic, we conclude that the tropical Atlantic remains

a challenging region for ocean models. High horizontal resolution has been suggested as a possible way to reduce biases, but we see in ICON-O and FESOM that even with 10 km horizontal resolution, biases remain large especially in the tropical Atlantic. While part of the tropical Atlantic model biases might be possible to address through tuning of the ocean models' vertical mixing parameterisation, the effect of this is highly model dependent and large biases remain even after tuning efforts. Due to the strong ocean-atmosphere coupling and feedbacks in the tropical Atlantic region, the effort to reduce the tropical

Atlantic model biases has to be addressed from both the atmospheric and oceanic model components as suggested by several other studies.

*Code and data availability.* The output of all model runs is provided for the upper 200m of the tropical Atlantic in the World Data Center for Climate (https://doi.org/10.26050/WDCC/nextGEMS_WP6oc, Bastin et al., 2023). The FESOM code of the version used is provided on Zenodo (https://doi.org/10.5281/zenodo.10617977, Danilov et al., 2024). The ICON-O code of the version used is provided on Edmond, the

Open Research Data Repository of the Max Planck Society (https://doi.org/10.17617/3.KUFQAM, Haak and Bastin, 2024).

## Appendix A: Bulk formulae

ICON-O uses two different set of bulk formulae depending on the forcing. In the case of ERA5 forcing, the bulk formulae from Kara et al. (2002) are used to compute the surface heat fluxes over the ocean and sea ice, with the water vapor pressure and 2 m specific humidity computed with (modified) equations from Buck (1981) and the longwave radiation following Berliand

(1952). The wind stress over ocean and ice is prescribed from ERA5.

Over open ocean, the turbulent heat fluxes are computed as follows. The enhancement factors (ratio of saturation vapor pressure of moist air to that of pure water vapor) at 2 m height ($f_a$) and at the ocean surface ($f_w$) are given by:

$$f_a = 1.0 + A_w + p_s(B_w + C_w T_d^2) \tag{A1}$$

$$f_w = 1.0 + A_w + p_s(B_w + C_w T_s^2) \tag{A2}$$

with $T_d$ the dew point temperature at 2 m height (in °C), $T_s$ the sea surface temperature (in °C), and the constants $A_w = 7.2 \cdot 10^{-4}$, $B_w = 3.20 \cdot 10^{-6}$, and $C_w = 5.9 \cdot 10^{-10}$. Then the vapor pressure at 2 m height ($e$) and at the water surface $e_w$ is computed as:

$$e = f_a a_w \exp((b_w - T_d/d_w)T_d/(T_d + c_w)) \tag{A3}$$

$$e_w = 0.9815 f_w a_w \exp((b_w - T_s/d_w)T_s/(T_s + c_w)), \tag{A4}$$

with $a_w = 611.21$, $b_w = 18.678$, $c_w = 257.14$, and $d_w = 234.5$.

The specific humidity at 2 m and at the ocean surface is then computed as:

$$q = \alpha e/(p_s - \beta e) \tag{A5}$$

$$q_w = \alpha e_w/(p_s - \beta e_w) \tag{A6}$$

with $\alpha = 0.62197$ and $\beta = 0.37803$. The relative humidity is computed as:

$$f = 0.39 - 0.05\sqrt{e/100}. \tag{A7}$$

The longwave radiation is computed following Berliand (1952). First, a factor for the effect from clouds on longwave radiation is derived:

$$f_c = 1.0 - (0.5 + 0.4/90 \min(|\Phi|, 60))CC^2, \tag{A8}$$

with $\Phi$ the latitude and $CC$ the cloud cover [0,1]. Then the net longwave radiation is computed as:

$$LW^* = f_c f \epsilon \sigma T_{2m}^4 - 4\epsilon \sigma T_{2m}^3 (T_s - T_{2m}) \tag{A9}$$

with $T_{2m}$ the 2 m temperature, $\sigma = 5.670 \cdot 10^{-8} \, \mathrm{W\,m^{-2}\,K^{-4}}$ the Stefan-Boltzman constant, $\epsilon = 0.996$ the emissivity factor.

The net shortwave radiation is computed as:

$$SW^* = (1 - \alpha_{vdi}) f_{vdi} I_0 + (1 - \alpha_{vdf}) f_{vdf} I_0 + (1 - \alpha_{ndi}) f_{ndi} I_0 + (1 - \alpha_{ndf}) f_{ndf} I_0 \tag{A10}$$

$I_0$ the incoming shortwave radiation, the factors for visible direct and diffusive radiation $f_{vdi} = 0.28$ and $f_{vdf} = 0.24$, and factors for near-infrared direct $f_{ndi} = 0.31$ and diffusive $f_{ndf} = 0.17$ radiation. The respective albedos are $\alpha_{vdi}$, $\alpha_{vdf}$, $\alpha_{ndi}$, and $\alpha_{ndf}$.

The bulk coefficients are then computed as follows. First the air density is computed as

$$\rho_a = p_s / (R_d T_{2m} (1.0 + 0.61q)), \tag{A11}$$

with $R_d = 287.04 \, \mathrm{J\,K^{-1}\,kg^{-1}}$ the gas constant for dry air. The 10 m wind speed is bounded by

$$U_{10} = \max(2.5, \min(32.5, U_{10})) \tag{A12}$$

Then the bulk coefficients are computed as

$$CD_1 = 10^{-3}(-0.0154 + 0.5698/U_{10} - 0.6743/(U_{10}^2)) \tag{A13}$$
$$CD_0 = 10^{-3}(0.8195 + 0.0506 U_{10} - 0.0009 U_{10}^2) \tag{A14}$$

The bulk coefficient for the turbulent latent heat flux is then computed as

$$CD_l = CD_0 + CD_1(T_s - T_{2m}) \tag{A15}$$

and bound to $CD_l = \max(0.5 \cdot 10^{-3}, \min(3.0 \cdot 10^{-3}, CD_l))$. The bulk coefficient for the turbulent sensible flux is computed as

$$CD_s = 0.95 CD_l \tag{A16}$$

The sensible $H$ and latent $E$ heat fluxes are then derived as

$$H = \rho_a c_p CD_s U_{10} f_r (T_{2m} - T_s) \tag{A17}$$
$$E = \rho_a L_f CD_l U_{10} f_r (q - q_w), \tag{A18}$$

with $f_r = 1.1925$ an energy budget closing factor for OMIP, $L_f = 2.5008 \cdot 10^6 \, \mathrm{J\,kg^{-1}}$ the latent heat of fusion, and $c_p = 1004.64 \, \mathrm{J\,K^{-1}\,kg^{-1}}$ the specific heat at constant pressure.

Over sea ice a similar approach is used, by first computing the enhancement factor

$$f_i = 1.0 + A_i + p_s(B_i + C_i T_i^2), \tag{A19}$$

with $A_i = 2.2 \cdot 10^{-4}$, $B_i = 3.83 \cdot 10^{-6}$, $C_i = 6.4 \cdot 10^{-10}$ and the ice surface temperature $T_i$. Then the vapor pressure and specific humidity over sea ice are computed as

$$e_i = f_i a_i \exp((b_i - T_i/d_i)T_i/(T_i + c_i)) \tag{A20}$$

$$q_i = \alpha e_i / (p_s - \beta e_i) \tag{A21}$$

with $a_i = 611.15$, $b_i = 23.036$, $c_i = 279.82$, and $d_i = 333.7$.

The bulk coefficients are then computed for the latent heat flux as

$$CD_{li} = CD_0 + CD_1(T_i - T_{2m}) \tag{A22}$$

bound to $CD_{li} = \max(0.5 \cdot 10^{-3}, \min(3.0 \cdot 10^{-3}, CD_{li}))$, and for the sensible heat flux as

$$CD_{si} = 0.95 CD_{li} \tag{A23}$$

The turbulent fluxes for latent and sensible heat are then derived as

$$H_i = \rho_a c_p CD_{si} U_{10} f_r (T_{2m} - T_i) \tag{A24}$$

$$E_i = \rho_a L_f CD_{li} U_{10} f_r (q - q_i) \tag{A25}$$

$$\tag{A26}$$

The net longwave and shortwave radiation are finally computed as

$$LW_i^* = f_c f_i \epsilon \sigma T_{2m}^4 - 4\epsilon \sigma T_{2m}^3 (T_i - T_{2m}) \tag{A27}$$

$$SW_i^* = (1 - \alpha_{vdi})f_{vdi}I_0 + (1 - \alpha_{vdf})f_{vdf}I_0 + (1 - \alpha_{ndi})f_{ndi}I_0 + (1 - \alpha_{ndf})f_{ndf}I_0 \tag{A28}$$

*Author contributions.* SB led the writing of the manuscript, ran the ICON-O model runs, and did the main analysis on the mean state. AK ran the FESOM model runs and wrote the parts about the FESOM model. FS analysed the diurnal cycle and wrote the corresponding text. OG implemented needed features in ICON-O and wrote the text about the bulk formulae. MAM did the analysis and wrote the parts on NIWs. TF analysed the deep cycle turbulence and wrote the corresponding text. RS contributed to analysis of the diurnal cycle. NK and HH assisted with the model runs. JJ, SD, MD and MJ conceptualised the idea and aquired funding. All authors contributed to initial analysis, planning and review of the manuscript.

*Competing interests.* The authors declare that they have no competing interests.

*Acknowledgements.* NextGEMS is funded through the European Union's Horizon 2020 research and innovation programme under the grant

agreement No. 101003470. The work was conceived of during a hackathon on the island of Bornoe in 2022: https://nbiocean.bitbucket.
io/bornoe2022a/) Florian Schütte is funded by the Bundesministerium für Bildung und Forschung as part of the NextG-Climate Science-
EUREC4A-OA project.

The vmADCP shipboard data are accessible at https://doi.pangaea.de/10.1594/PANGAEA.899052 (Burmeister et al., 2019).

The Argo float data were collected and made freely available by the International Argo Program and the national programs that contribute

to it. (https://argo.ucsd.edu, https://www.ocean-ops.org). The Argo Program is part of the Global Ocean Observing System.

The authors thank the German Climate Computing Center (DKRZ) for access to their computing infrastructure.

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

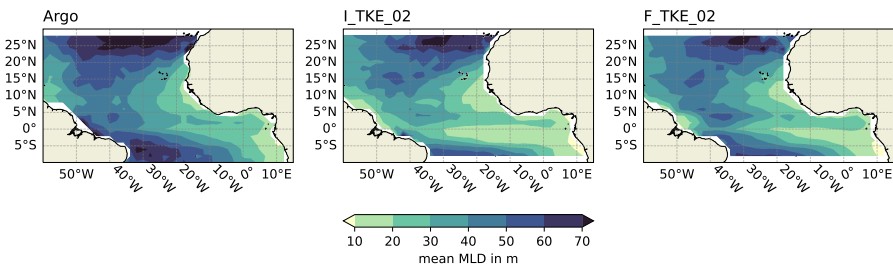

**Figure 1.** Annual mean mixed layer depth (MLD) in the tropical Atlantic Ocean from Argo float data (Argo, 2022, mean over entire available measurement period from 2000 to 2022) and sensitivity model runs from FESOM and ICON-O (mean over second integration year 2015). The MLD has been calculated using a commonly used density threshold criterion, where the MLD is defined as the depth at which the potential density exceeds the potential density at 5 m depth by more than 0.125 kg m$^3$.

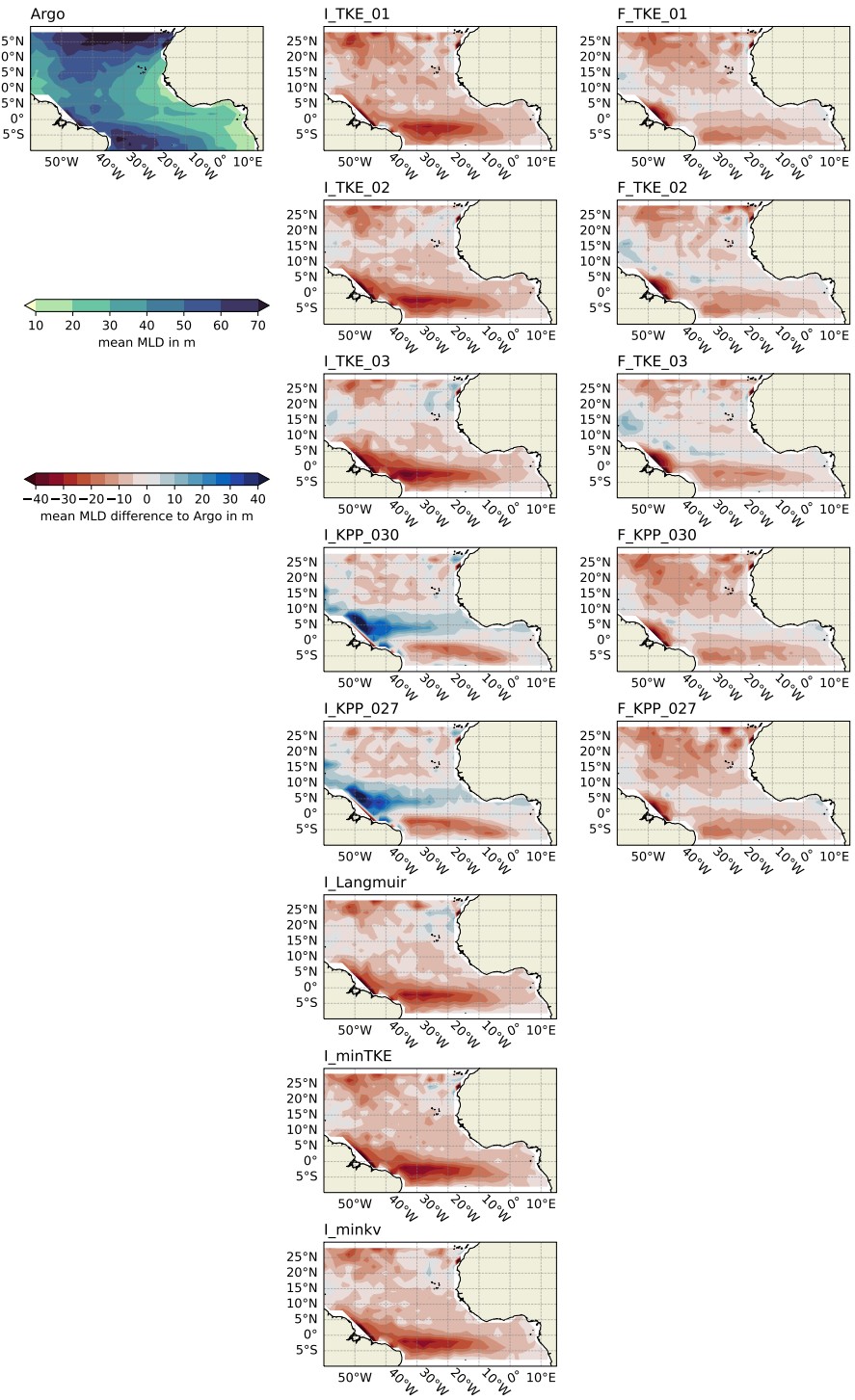

**Figure 2.** Annual mean mixed layer depth in 2015 for the different simulations of ICON-O and FESOM (centre and right panels) relative to the Argo climatology (top left panel).

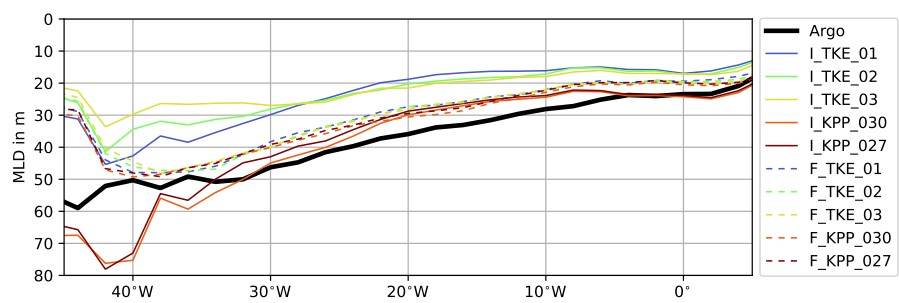

**Figure 3.** Annual mean Atlantic Ocean mixed layer depth along the equator (averaged between 4°S and 4°N) from Argo float data as well as FESOM and ICON-O.

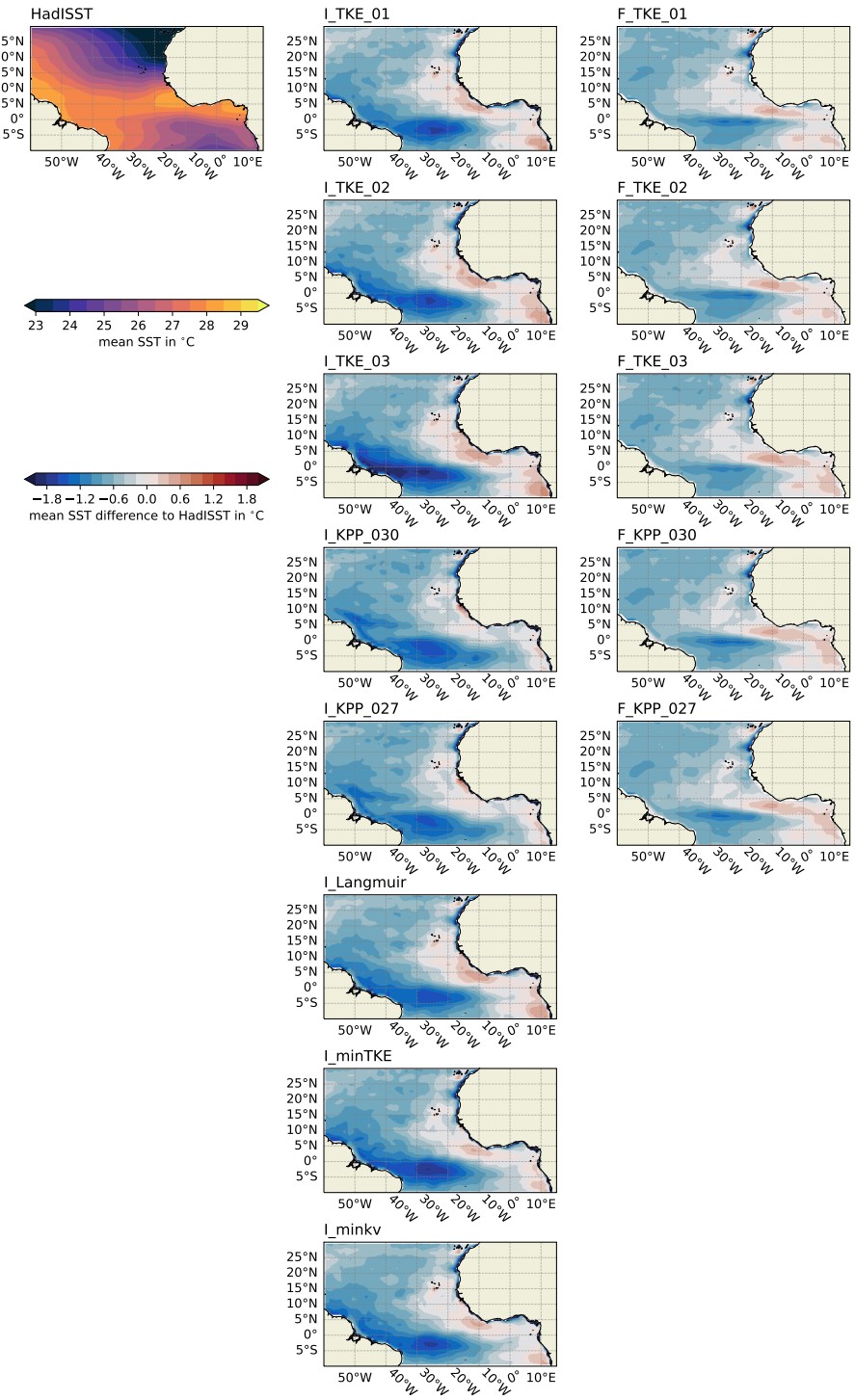

**Figure 4.** Annual mean 2015 Atlantic Ocean sea surface temperature from the HadISST dataset as well as FESOM and ICON-O (for the models, differences to HadISST are shown, blue colours meaning the model is colder than HadISST).

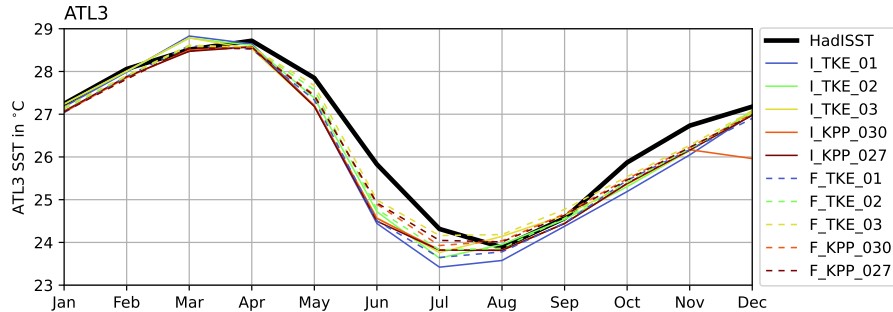

**Figure 5.** Seasonal cycle of sea surface temperature averaged over the ATL3 box (20°W-0°E, 3°S-3°N), from HadISST data as well as FESOM and ICON-O.

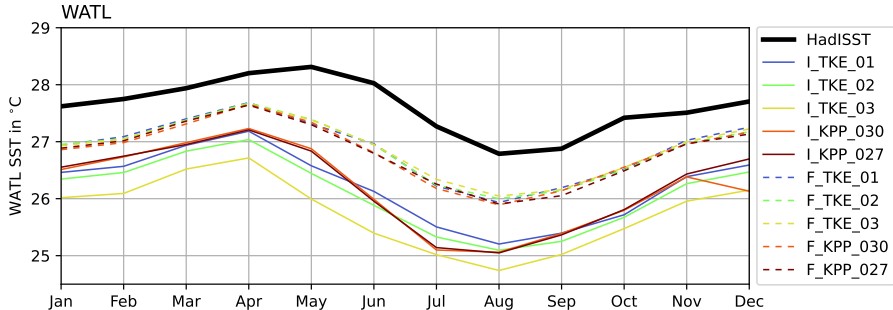

**Figure 6.** Seasonal cycle of sea surface temperature averaged over the WATL box (45°W-25°W, 3°S-3°N), from HadISST data as well as FESOM and ICON-O.

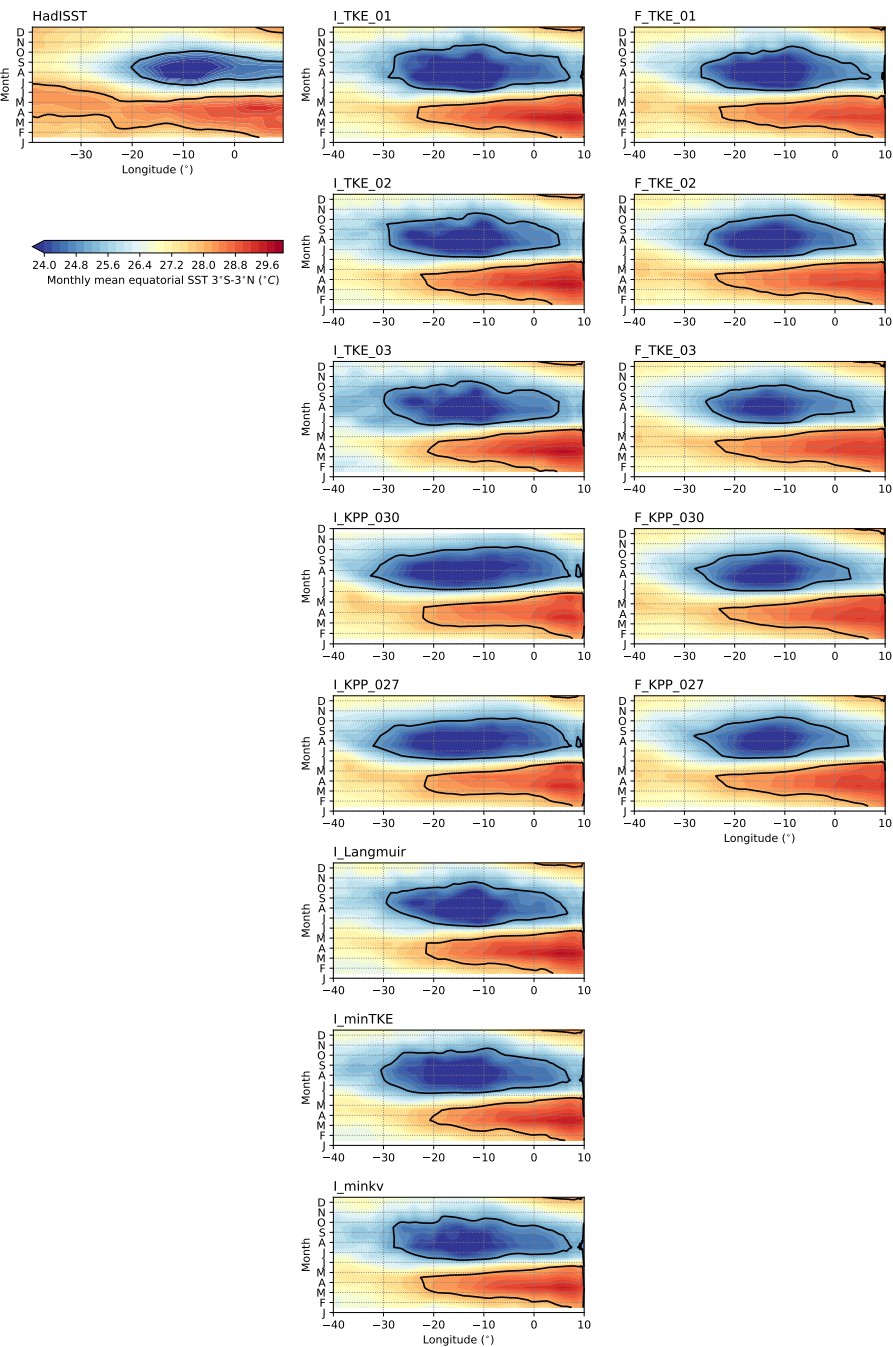

**Figure 7.** Hovmoeller plots of monthly mean SST in 2015 averaged between 3°S and 3°N for the HadISST dataset (left), the ICON-O simulations (center) and the FESOM simulations (right). The black contours correspond to 25 and 28°C.

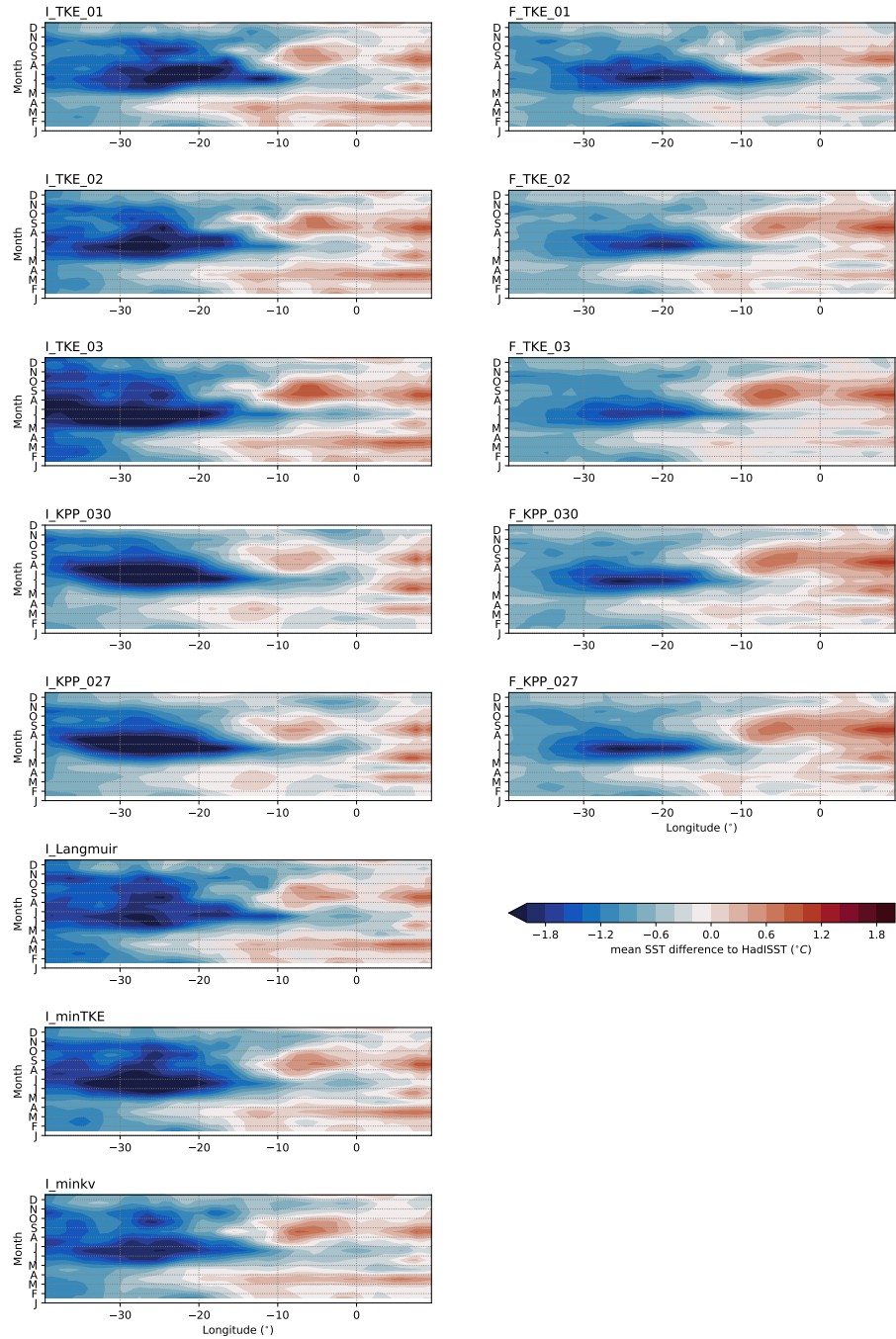

**Figure 8.** Hovmoeller plots of the SST bias for ICON-O (left) and FESOM (right). The SST bias is computed relative to the HadISST data from 2015 and uses the same monthly mean SST shown in Figure 7.

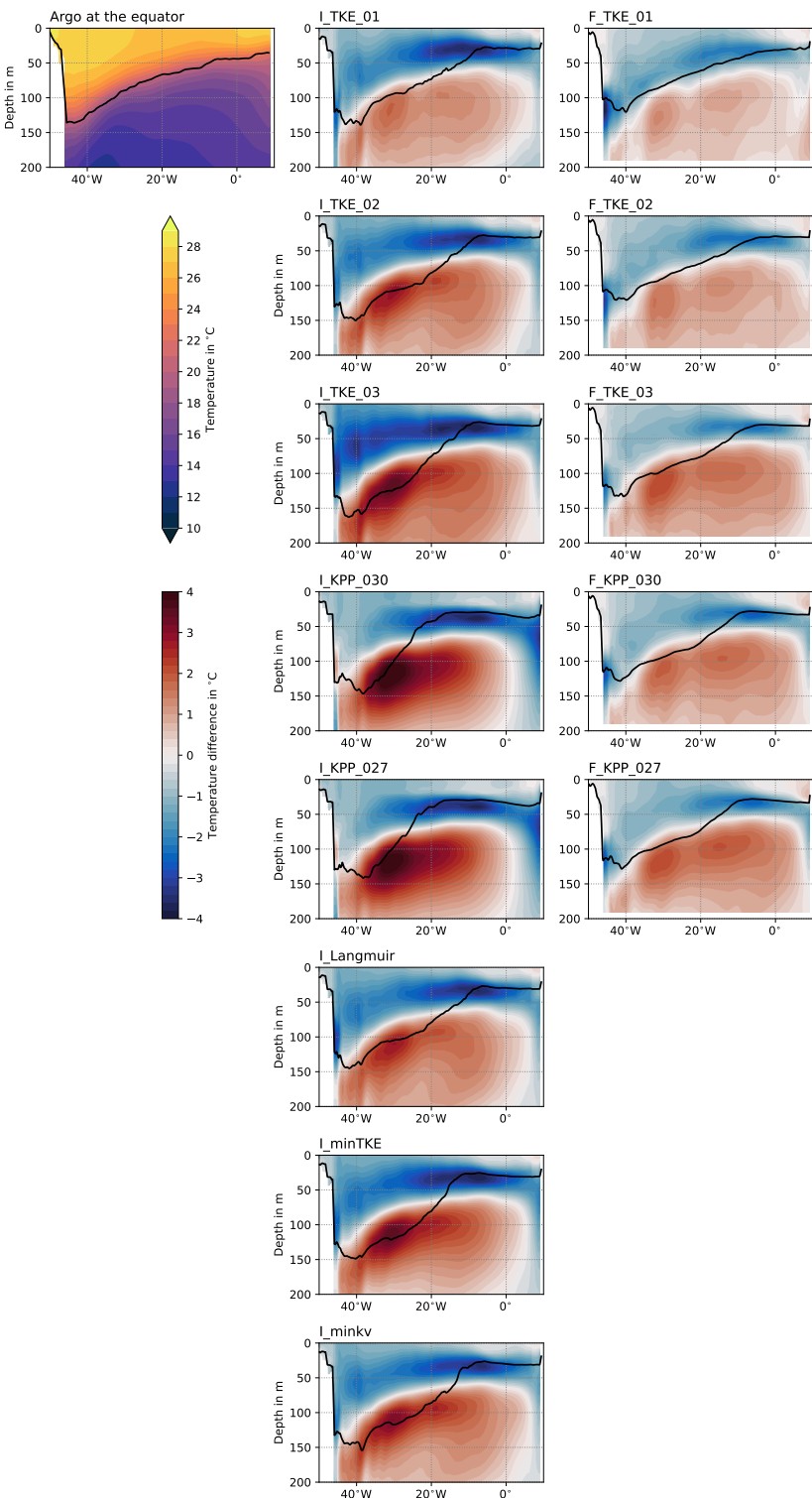

**Figure 9.** Vertical section along the equator of 2015 annual mean temperature (shading) and the 2015 annual mean thermocline depth (black contour) for Argo float data (top left panel), and difference to Argo for FESOM and ICON-O. The temperature sections have been averaged between 1°S and 1°N.

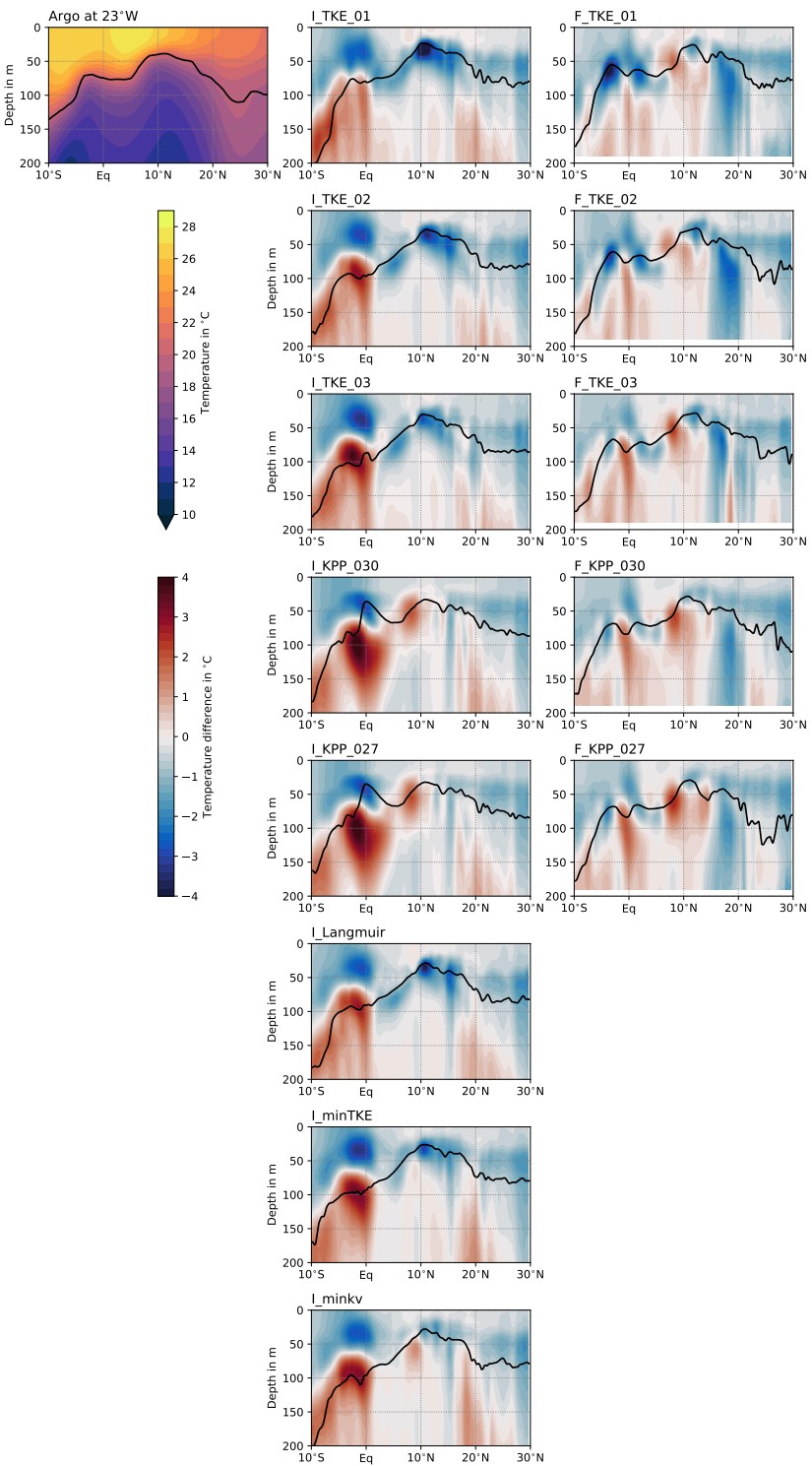

**Figure 10.** Vertical section along 23°W of 2015 annual mean temperature (shading) and the 2015 annual mean thermocline depth (black contour) for Argo float data (top left panel), and difference to Argo for FESOM and ICON-O. The temperature sections have been averaged between 24°W and 22°W.

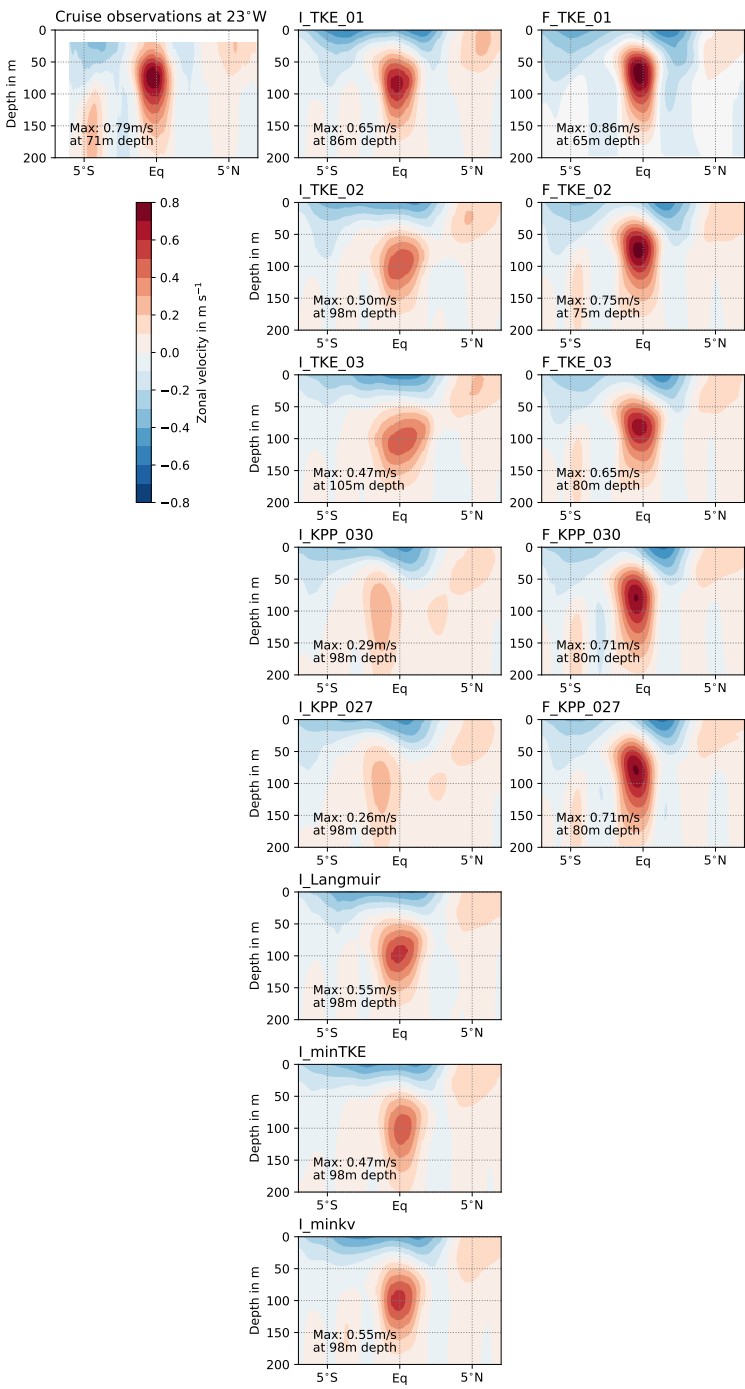

**Figure 11.** Strength of the Equatorial Undercurrent (EUC) from cruise observations and FESOM and ICON-O model runs. Shown is a mean section of zonal velocity along 23°W, where multi-year cruise data are available. The observations are averaged over all available years, the sections from the models are annual averages over 2015. The model velocity sections are averaged between 24°W and 22°W.

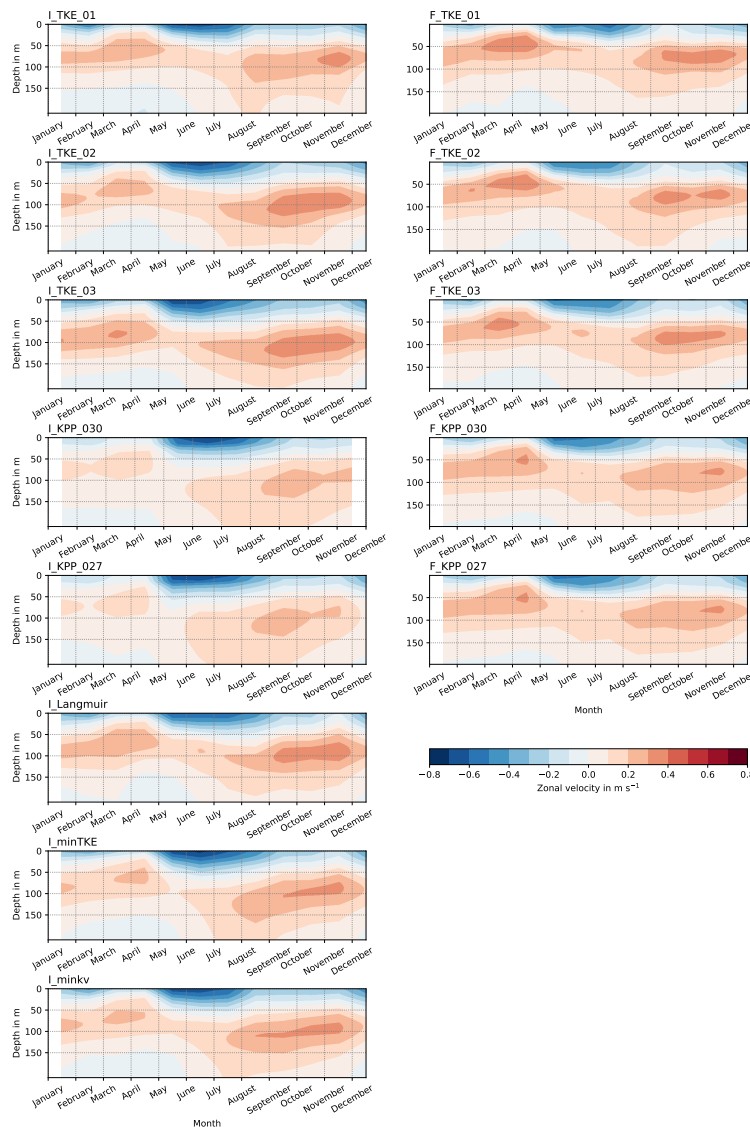

**Figure 12.** Hovmoeller plot of monthly-mean zonal velocity in 2015 as a function of depth at 23°W, where the cruise observations displayed in Figure 11 are located, averaged between 3°S and 3°N. The panels on the left are for the various ICON-O simulations and those on right for the FESOM simulations.

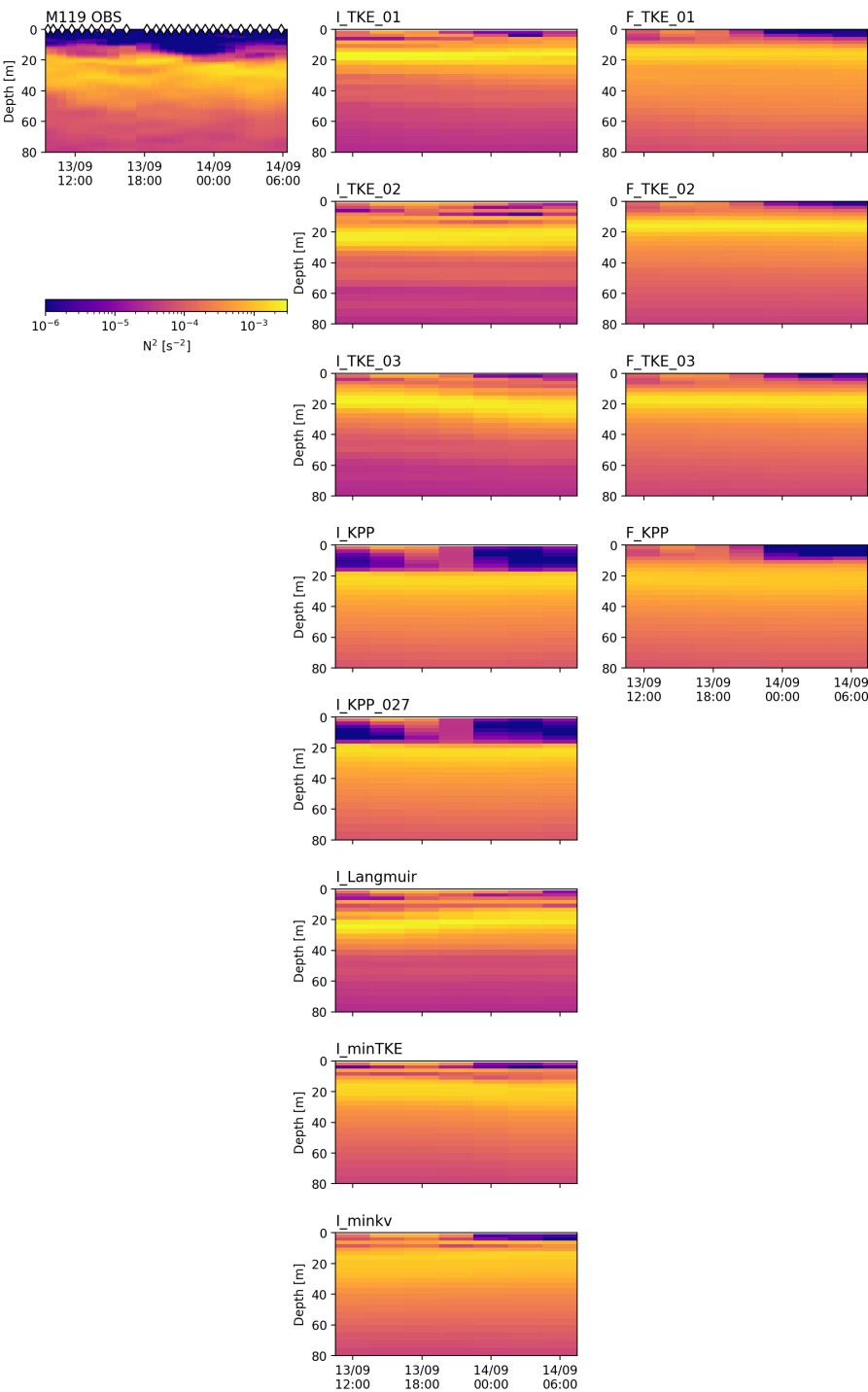

**Figure 13.** Buoyancy frequency ($N^2$) in shipboard observations and the models at 11°N, 21°W between the 13th and 14th of September, 2015. From the models data from one grid point at the given location is shown, to directly compare to the M119 observations at the same location. The diamonds in the subfigure showing M119 observations correspond to the points in time at which the measurements were taken.

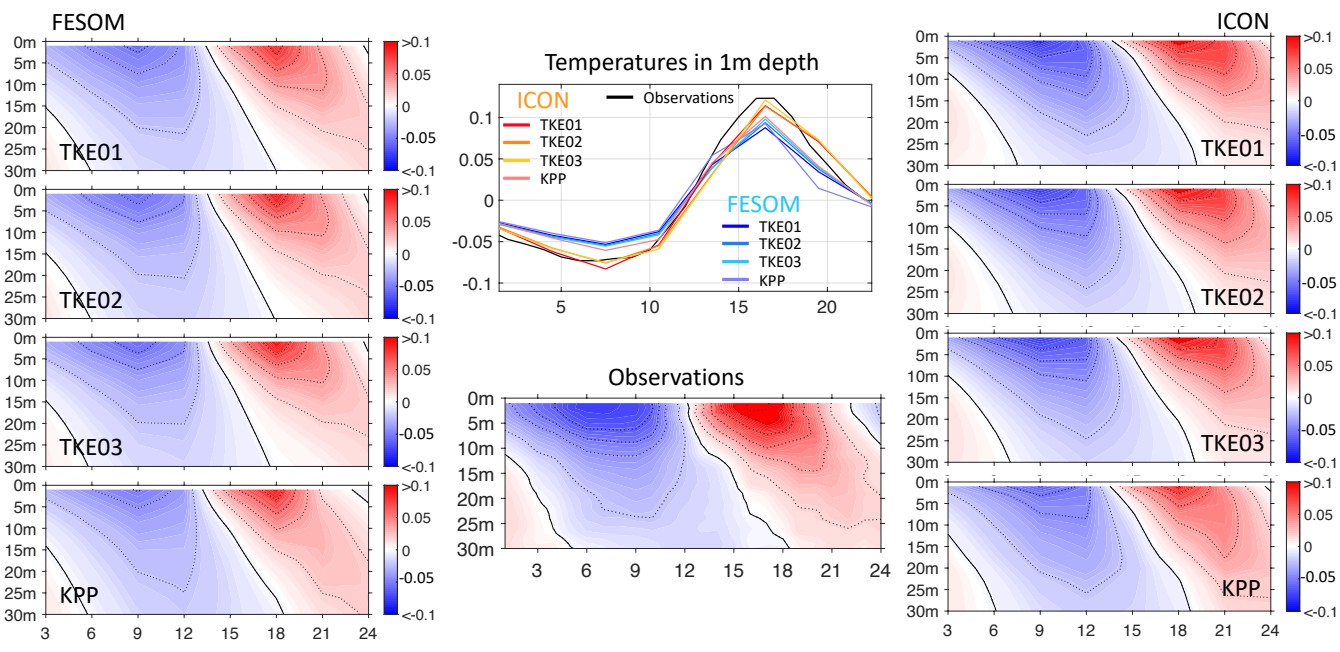

**Figure 14.** Composite diurnal cycle of the upper-ocean daily temperature anomaly (FESOM - left ; Observations - bottom center; ICON - right). In the top center the daily anomalies at 1 m depth are shown (FESOM - blue; Observations - black; ICON - red).

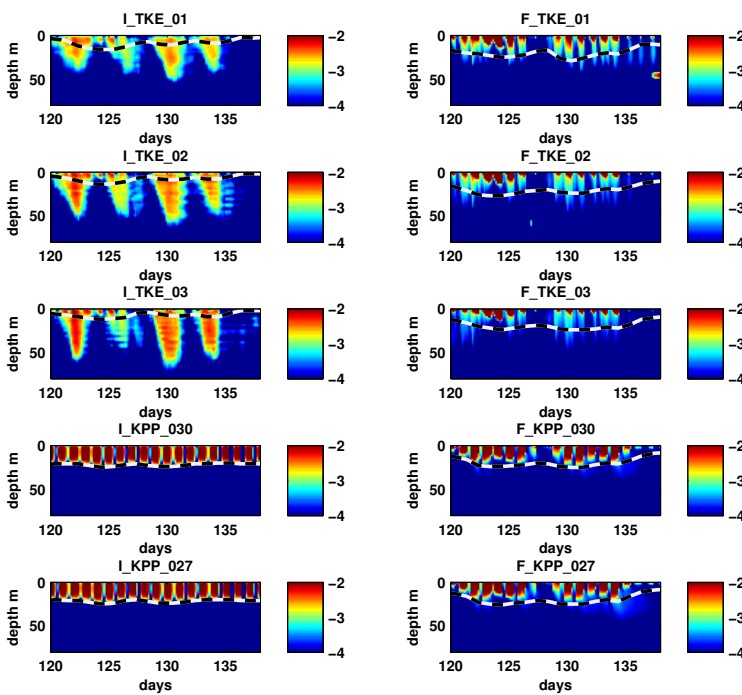

**Figure 15.** Example timeseries of vertical eddy diffusivity $k_\nu$ at 0°N, 23°W for days 120 to 138 in 2014 illustrating the different characteristics of DC turbulence in the model runs. Three main groups can be distinguished. ICON TKE runs capture the downward propagation of the DC turbulence to observed depths but not its frequency, FESOM TKE runs exhibit weak diurnal varying mixing below the MLD, while KPP runs show elevated mixing confined to the mixed layer, only. Color contours are $\log_{10}(k_\nu)$ in $\mathrm{m^2 s^{-1}}$. Dashed lines indicate the depth of the night time mixed layer.

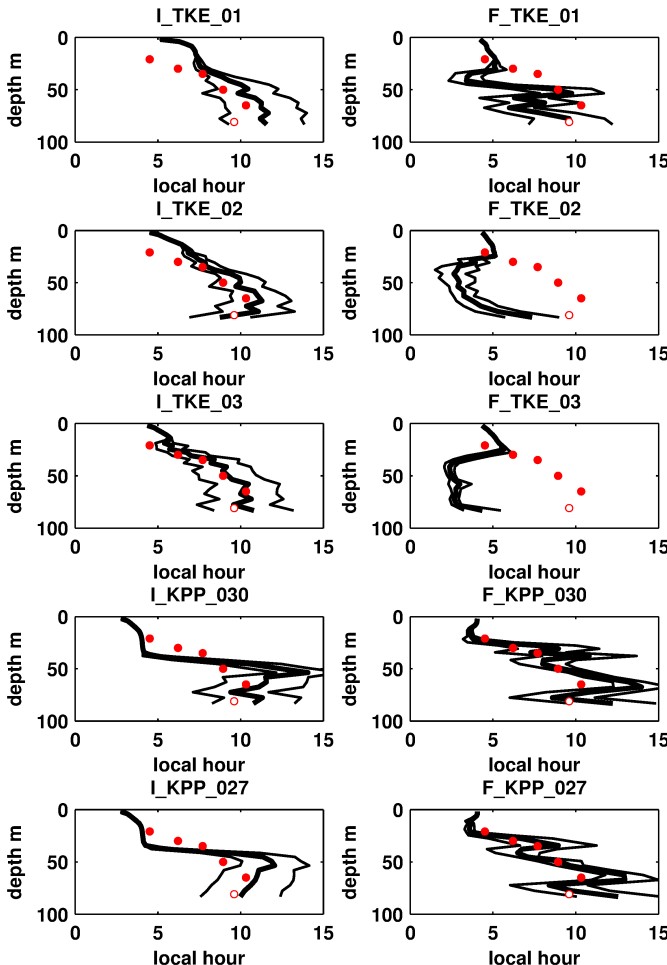

**Figure 16.** Average local daytime of maximum diffusivity as function of depth for the different model runs in black, and from PIRATA chipod observations in red, at 0°N and 23°W. Uncertainty ranges are the standard deviation from all estimates in the area [0.5°S, 0.5°N, 23.5°W, 22.5°W].

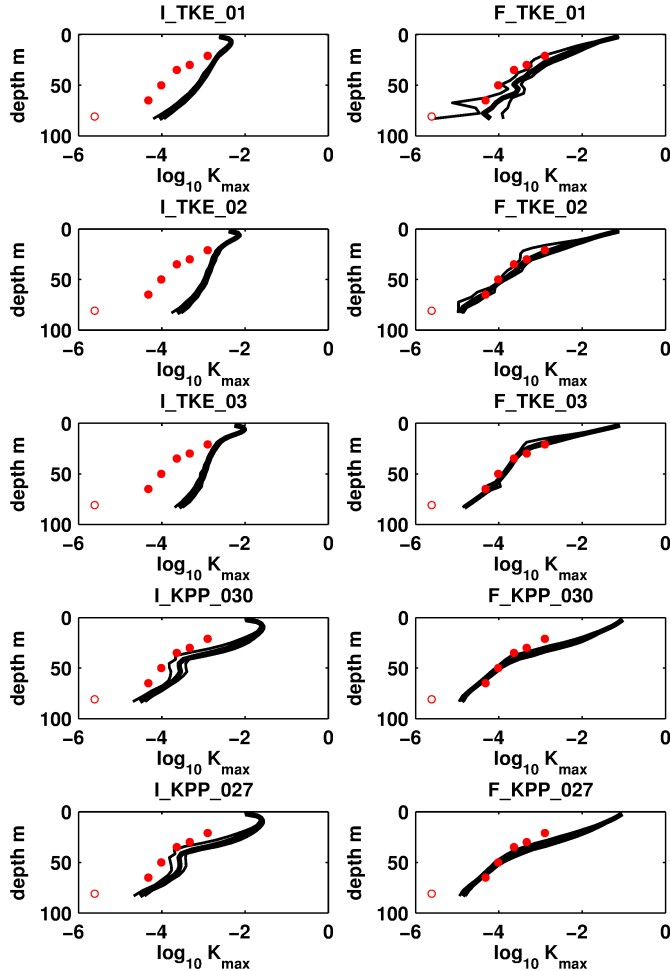

**Figure 17.** Average maximum diffusivity as function of depth for the different model runs in black, and from PIRATA chipod observations in red, at 0°N and 23°W. Uncertainty ranges are the standard deviation from all estimates in the area [0.5°S, 0.5°N, 23.5°W, 22.5°W].

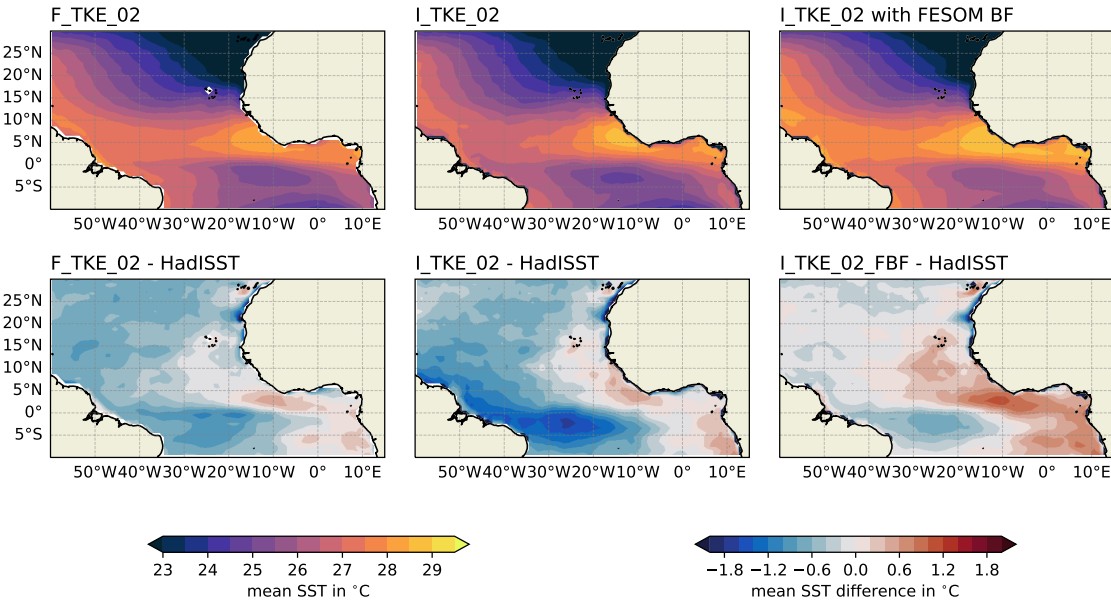

**Figure 18.** Effect of exchanging the forcing bulk formulae in ICON-O on the 2015 annual mean sea surface temperature. The upper three panels show the 2015 annual mean SST from the FESOM TKE_02 run and the ICON-O TKE_02 run as well as the ICON-O TKE_02 run with the bulk formulae used in FESOM. The lower panels show from the same three runs the difference to 2015 annual mean SST from HadISST as shown in Figure 4.

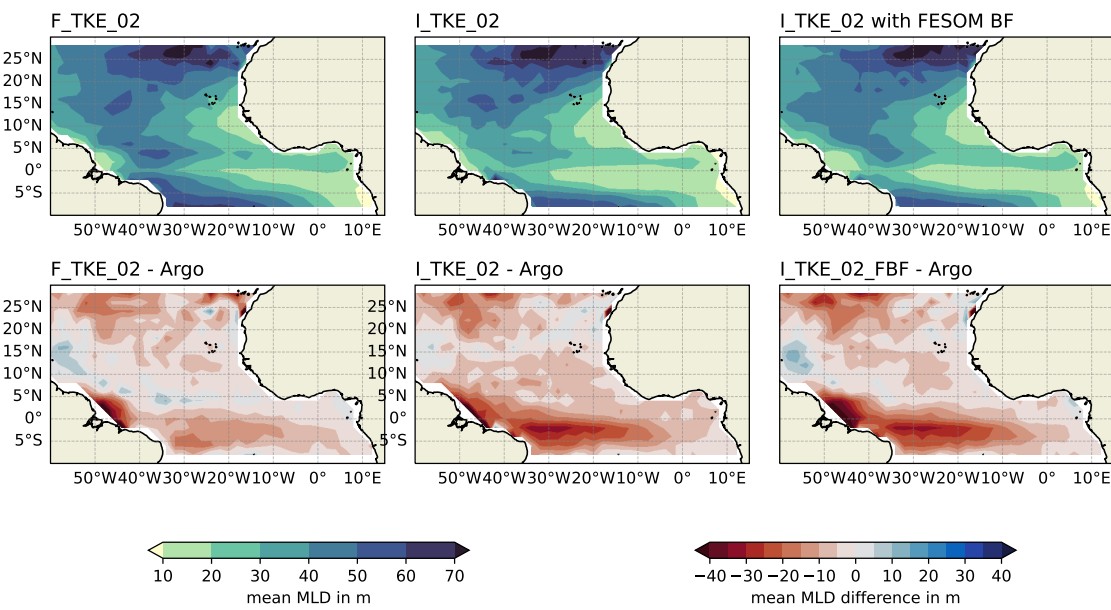

**Figure 19.** Effect of exchanging the forcing bulk formulae in ICON-O on the 2015 annual mean mixed layer depth. The upper three panels show the 2015 annual mean MLD from the FESOM TKE_02 run and the ICON-O TKE_02 run as well as the ICON-O TKE_02 run with the bulk formulae used in FESOM. The lower panels show from the same three runs the difference to the mean MLD from Argo float data as shown in Figure 2.

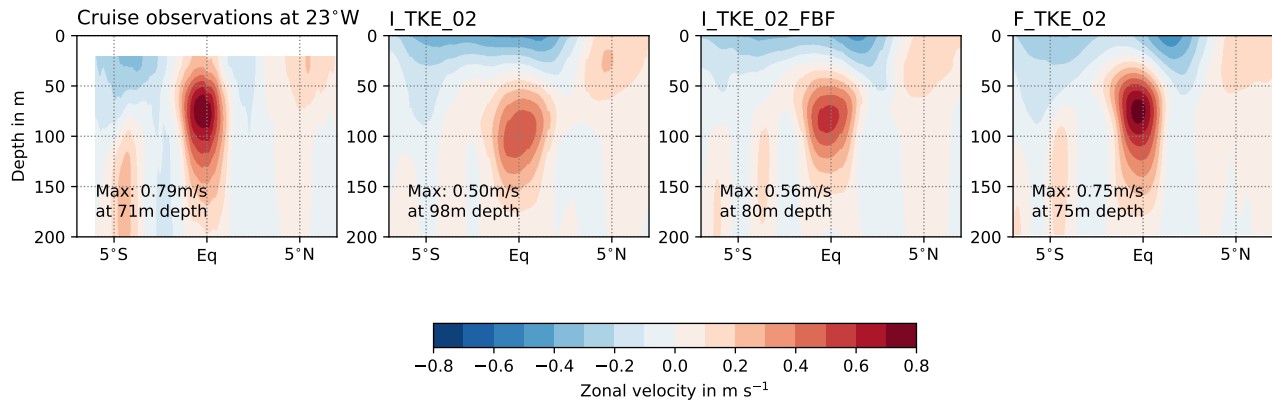

**Figure 20.** Effect of exchanging the forcing bulk formulae in ICON-O on the Atlantic Equatorial Undercurrent. Shown is the 2015 annual mean zonal velocity along 23°W, from cruise observations, from the ICON-O TKE_02 run, from the ICON-O TKE_02 run with the bulk formulae used in FESOM, as well as from the FESOM TKE_02 run.