# Peer review of "Sensitivity of the tropical Atlantic to vertical mixing in two ocean models (ICON-O v2.6.6 and FESOM v2.5)"

_EGUsphere, 2024_

## Author Response (AR1)

**Author Response to reviewers' comments on Bastin et al. (EGUSPHERE-2024-2281)**

December 13, 2024

**General remarks: All three posted comments (CC1, RC1, RC2) are answered below in three separate sections. The referees'/commenters' original text is quoted as regular text, while our point-by-point replies are marked in bold. The line numbers given in our replies refer to the revised manuscript pdf with marked changes.**

**1 CC1**

The paper investigates the impact of turbulence parameterizations in ocean models, focusing on the equatorial Atlantic in 2015. Two ocean models that are also part of climate models are considered: ICON-O and FESOM. They both have intermediate high horizontal and vertical resolution (128 layers), but with very different horizontal grids and schemes. Both have a z* vertical grid (with SSH link, but not quite the same, if I correctly understood). However, they have rather comparable near-equatorial horizontal resolution. These are forced runs (ERA5 forcing terms, mostly) with use of bulk formula for the air-sea exchanges. Runs are typically done on two years: different turbulent schemes are tested (in particular In ICON-O) as well as the bulk formula formulation (to test the influence on the runs of the differences in flux formulation between the two). The test runs are two years long, the second year been considered, which should be enough for the near equatorial adjustment, but avoid the larger basin scale adjustments that will result from the different turbulence parameterizations. Notice also that most of the changes made should mostly modify the near surface mixing, and not so 'directly' the deeper one. The bulk formulas used both imply negative feedback towards the ERA5 atmosphere temperature in the tropical Atlantic. However, in regions where the model produce excess surface temperature (such as south-eastern equatorial Atlantic), either because of the other components of the heat budget (radiative...) or because of the turbulence scheme or model simulations, this would add a destabilizing term in the mixed layer, and thus moderate excess near surface stratification. In regions, where the models are too cold (probably too much upwelling or thermocline structure not well reproduced), this would contribute to some added stratifying term, and thus very reduced MLD. I am just stating that as there could be a link between SST biases and MLD bias structures related to the overall bulk formulation (which ever of the two is used). The main results are that MLD underestimation bias (and SST too low bias) is overall large (almost a factor of two in some areas and runs; most noticeable between 0 and 10°S in central and western Atlantic), and although sensitive to the turbulence parameterization, not to the point of changing main patterns (same for the SST bias structure). There is some dependency nonetheless on scheme which is explained, and systematic differences remaining between the two models. As pointed out by the authors, some of this bias might be due to overall thermocline structure and flow. There is then ad good discussion on high frequency variability in particular the diurnal warm layer (but also inertial waves). This is likely important, due to modulation and possible impact on momentum flux in the ocean (less so for heat and water flux, as it is more linear...).

This made me wonder about the 'in situ' reference used for those. It is based on Argo data using (according to figure 1 caption) all Argo profiles in 2000 to 2022), which implies that a large part of the profiles were not starting above 7 m, and thus missing most of the DWL. There is also a question on when in the day the Argo profiles arrive at the sea surface, which is often not homogeneous across the tropical Atlantic or through the day. Are the authors sure that there is no systematic difference due

to that the in-situ Argo data reference, and by how much (and could there be a spatial structure in it due to distribution of timing of Argo profiles through the day). At least, this is not consistent with what is presented from the model, which uses a surface reference for density (a density criterion is used). In some ways, it could have been preferable to use for the models a late-night value to compare with the Argo climatology (and for those always taking the reference near 7 m, or other fixed depth).

**Thank you for pointing this out. We have changed the calculation of the mixed layer depth – we are now using 5 m depth as reference level instead of the surface, which should exclude most of the strong near-surface diurnal variability. We then compute the mixed layer depth as the depth where potential density exceeds the value at 5 m depth by 0.125 kg/m3 (criterion used in e.g. Levitus, 1982). It is recommended to compute MLD in OMIP and CMIP models using a threshold value of 0.03 kg m$^{-3}$ which we also used before (Griffies et al., 2016; see also the discussion by Treguier et al., 2023). However, in the tropics, this threshold corresponds to a temperature difference of less than 0.01 K, which may reflect MLD changes due to diurnal warming in the near-surface layer as you pointed out. The results stay qualitatively similar, suggesting that the analysis of the Argo data yields robust results. We have added these changes to the manuscript (see lines 222ff. in the revised manuscript with differences marked).**

Whereas the investigation of DWL and diurnal cycle require another analysis (as is done). For SST, it is HADISST which is used as a reference. It would be important to remind whether it is the daily average SST which is used for the comparison (or something else). There are also interesting results on near equatorial mixing and day time of maximum diffusivity, deep cycle turbulence, with suggestions of some of the TKE (or PKK) runs performing more satisfactorily than the others (for these investigations, other data sets are used, which seem appropriate for the investigation, as well as fr the off equatorial DWL) (well for deep-cycle turbulence it is less so, based on figure 13 and 14) Overall, would it be fair to say that somehow, we have two models with rather large systematic large-scale biases (as also seen in Figure 7) that would not change much in such a short term as 1 to 2 years of the tests, which mostly tackle the surface mixing (although some minimum values also impact the subsurface terms). Maybe that could be a reason with the differences in the overall results with what is found in Deppenmeier et al (2020) investigating ck dependency of bias in NEMO (larger ck leading to surface cooling and subsurface warming, and less SST bias).

**We agree that two years is a short duration of our model runs. Unfortunately we cannot run the models longer due to limited computing resources. However, the runs are spun up before we change the parameters in the mixing schemes in the beginning of 2014. The effect of the parameter changes on the surface mixed layer and the near-surface processes in the tropical Atlantic should then be in place much faster than after one year. We have indicate this in the manuscript, both in the model description and in the discussion in connection with the comparison to other studies as suggested (lines 148ff., 577ff. in the revised manuscript with differences marked).**

Alltogether it is a rather interesting study worth publishing.

**Thank you.**

Minor comments.

1. For TKE they use Pr=6.6Ri which I find large (I believe that it is 1 in NEMO, but have not checked). Why this choice?
   **We use this formulation because it is the default value in ICON-O and has been found to give sensible values for the mixing efficiency. The turbulent Prandtl number is also capped so that it always stays between 1 and 10.**

2. l. 372: 'to a large extent on the surface velocity of the ocean' ('to a large extent' may be a little too strong; more for energy than for heat/water). After, I understood the point made on

the relative direction of wind and currents, and thus the difference between the equatorial and off-equatorial situations, but there to impact larger for energy/wind power than for heat/water (and it is not so clearly separated, according to a recent paper, Hans et al (2024)) The authors are likely aware of the Hans et al. paper (JGR Oceans in Press) that carefully evaluates from data the structure of the DWL and its diurnal jet along the equatorial Atlantic, and could complement what is discussed in the paper.

**Thank you for the feedback. The three sentences starting at line 370 were indeed not correctly formulated. The influence of the diurnal jet on wind power input is significant due to the scalar product of water and wind velocities. However, its effect on heat fluxes is somewhat smaller. Of course, it is still present due to the reduced wind stress and the other energy input (allowing for better downward mixing of heat), but it is not as pronounced as previously stated. We hope the revised formulation clarifies this point: "This diurnal jet of surface water influences wind stress and wind power input, thereby affecting the exchange of properties such as momentum, moisture, and heat between the ocean and the atmosphere. Air-sea fluxes are partially dependent on the surface water velocity aligned with the wind direction, which itself is influenced by the stratification caused by the diurnal warm layer (DWL). When the surface flow deviates from the wind direction due to Coriolis deflection, the impact on air-sea fluxes diminishes." (lines 451ff. in the revised manuscript with differences marked.)**

3. Figure 2: I would write instead: "Annual mean mixed layer depth in 2015 (correct?) for the different simulations of FESOM and ICON-O relative to the Argo climatology (for 2000-2002?) presented on the left top panel. (or difference in annual MLD..., but not 'between FESOM and ICON-O runs')

   **We have changed this as suggested (see caption of Figure 2).**

4. Captions of figure 7 and 8 incorrect. The panels show SST difference (except for the Argo one).

   **We have changed this (see captions of Figures 9 and 10 in the revised manuscript).**

5. Figure 7 caption: over which latitudinal band are the 2015 Argo profiles (the reference for the other panels) averaged. Does this averaging scale have an impact (or not) on the anomalies presented on the other panels for the different model runs. Altogether, some figure captions are not very detailed (and information has to be retrieved from the core of the paper to figure to what they correspond. Another example is figure 11, in which no special domain is specified for the model runs, nor where the observations were collected.

   **We have added the information to the caption of Figure 7 (now Figure 9 in the revised manuscript), and have also tried to make the other figure captions more detailed. (See e.g. captions of Figures 9 and 14 (formerly Figure 11).)**

6. Figure 10, I understand what is attempted, but I have a hard time looking at it, convincing myself on what is said in the paper. In this case, is it important to show all the panels. I can imagine many reasons that may not be that relevant for the overall conclusions, why the model runs don't reproduce the special event found in the data.

   **We are not sure how we can improve this. We will keep all the panels of the figure.**

7. Figure 12: I understood afterwards the choice of days 120-138 of year 2014 (reading the paper). On the other hand, the dates are rather close to the beginning of the test simulations, and could be sensitive to it. The results are very different between the runs, for example the lower (and not daily?) modulation in I KPP 01 to 03. I did not fully understand what is from it the lesson. Why is there this 5-day modulation in these three runs and not in the others; Would they have had some 'instability' waves, for example, that are not present in the other runs. And from those panels, how does one feel what is expected? (the last sentence of the caption is for that a bit vague, and not informative)

   **We have rewritten much of the section describing the DC turbulence results and hope to have improved its clarity and lessons learned (see lines 473 ff. in the revised manuscript with differences marked). We have also tried to improve the caption of Figure 12 (now Figure 15 in the revised manuscript). As for the longer-than-diurnal**

**turbulence cycle that is visible in the ICON-O runs, we are not sure what causes this. We suspect that the turbulence in the thermocline does not dissipate fast enough in the mornings and/or still draws energy from the current shear during the day.**

8. Fig. 15: Is what is shown the results in ICON-O of using the alternative bulk formula (and with comparison to the observed MLD, as in Fig. 2). Or is it instead the difference of the two sets of runs which the caption would suggest.

   **It is the results in ICON-O of using the alternative bulk formula (and with comparison to the observed MLD, as in Fig. 2). We have made the figure caption (also that of the following two figures) more detailed to make this clear (see captions of Figures 18, 19, 20 in the revised manuscript).**

9. Figure 17: I assume that only the third panel from left with the alternate forcing bulk formulae in ICON-O. I would remove in the title 'Effect of exchanging'... and be more specific on what are the runs presented...

   **We have made the figure caption more detailed (see caption of Figure 20 in the revised manuscript).**

**2 RC1: Anonymous Referee**

**2.1 Summary**

Using OMIP-type simulations with two OGCMs, ICON-O and FESOM, the authors investigate the influence of the vertical mixing parameterization on the simulation of the tropical Atlantic, with both the KPP and TKE schemes tested with several parameter settings. While the models do show sensitivity to the choice of mixing scheme and parameter setting, the choice of model has a larger impact. Furthermore, all simulations show relatively similar bias patterns, including a weaker than observed equatorial SST gradient, which suggests that the vertical mixing scheme may not be the major cause of these biases. The manuscript is well written and presents a thorough analysis of the performance of the two models. One concern I have is that one year of simulation (the year 2015) may not be sufficient to reliably assess model performance. I presume the high-resolution simulations are too expensive to conduct long-term integrations, but in that case one may wonder if simulations with lower resolution but longer integration time would have been more suitable. Itemized comments follow below.

**We would like to thank the reviewer for their thoughtful and constructive feedback. In this study, we choose to focus on high-resolution resolutions because of the presumed importance of the mesoscale for mean state of the tropical Atlantic climate (Seo et al., 2007) and reducing tropical biases (Small et al., 2014). The reviewer rightly points out that such high-resolutions are computationally too expensive to run for longer integration periods. Furthermore, storing high frequency output data (3-hourly) required for the analysis in the paper would be challenging for longer integration periods. Nevertheless, we agree that the short integration length is a limitation of the study and outline below how we will address this limitation in the revised paper draft.**

**2.2 Major Comments**

1. It is not clear whether one year is long enough to reliably assess model performance. Uncertainty in both the observations and the single-realization simulations could be comparable to the biases you are trying to examine. If long-term OMIP simulations at lower resolution are available for ICON-O and FESOM you could check how representative a single year is by comparing the bias of individual years with that of the long-term mean. Alternatively, you could also look at OMIP simulations from other modeling centers to get a rough idea.

   **Thank you for the suggestion. We agree that two years of the sensitivity runs (of which we analysed the second year) is not a long time to assess model performance. Unfortunately, we cannot run the models for longer because of limited computing resources. However, we would argue that the main point of the manuscript is not**

to check model performance, but to assess the effect of changing the vertical mixing parameters on model performance. Since we looked at the same year in all sensitivity runs and they were forced ocean-only runs, we think that the comparison between the sensitivity runs is valid. The adjustment of the upper ocean to the changes in the mixing parameters should happen on a time scale much less than a year, so that the data from 2015 (the second year of our integrations) should be useable for our purpose. We still agree with you that it is important to check how representative a single year is compared to the long-term mean. For the ICON simulation, we have a spin up simulation for the period 2010-2021, from which the sensitivity runs were initialised in 2014. In Figure 1 in this reply below, we have plotted the biases for annual mean SST with respect to the HadISST dataset. As you can see, the cold bias in the western tropical Atlantic is somewhat stronger in 2015 than the other years shown. However, the interannual variability in ICON-O is smaller than the bias itself and does not change the large scale bias pattern, which stays very similar over all the years shown in the figure. Since the bias pattern is consistent over the different years of the ICON-O model simulation, we would argue that the output from the year 2015 can be used to assess the effect of changes in the mixing scheme on the equatorial Atlantic biases, even though it is a relatively short time span and the cold SST bias in that year in ICON-O is larger than in other years. For FESOM, the output data for the corresponding spin up is unfortunately no longer available. However, we can compute the SST biases for a simulation with comparable grid spacing in the tropical Atlantic, as shown in Figure 2 in this reply. Similar to the ICON spin up run, we see that the magnitude of the interannual variability of the SST biases is smaller than the magnitude of the SST biases themselves. As mentioned in the manuscript, the year 2015 in particular was chosen for the sensitivity experiments because from that year, there are observations of near-inertial waves that we use to validate the model results. We have added some discussion on this to the manuscript (see lines 148ff. and 577ff. in the revised manuscript with differences marked).**

2. In the introduction, you refer to previous studies suggesting that the biases in CGCMs and their corresponding OMIP simulations are similar. But the biases you show in the ATL3 region are actually rather atypical, with temperatures lower than observed and the minimum occurring one month early. In many CMIP6 models, on the other hand, the SSTs in the ATL3 are too warm and the minimum is reached too late. The discrepancy could be due to the high model resolution, as stated by the authors, but this cannot be assessed without a corresponding CGCM simulation. Do the authors have such simulations available?

   **Unfortunately we do not have corresponding CGCM simulations available. Instead, we compare the temperature biases from the ocean-only ICON-O and FESOM simulations to the biases found for OMIP2, CMIP6 and HighResMIP in the studies by Richter and Tokinaga (2020) and Farneti et al. (2022). They find that for High-ResMIP, the warm SST bias in the eastern tropical Atlantic is reduced compared to CMIP6, which would fit to our guess that this might also be the case in our model runs because of the high horizontal resolution. Farneti et al. (2022) also show the subsurface temperature bias along the equator for OMIP2, CMIP6 and HighResMIP (their Figure 6). The equatorial subsurface temperature bias in our ICON-O and FESOM runs are rather untypical compared to the OMIP2 multi model mean bias, which is too warm in the upper 200 m. The bias in ICON-O and FESOM rather resemble the multi-model mean subsurface temperature bias from CMIP6 and HighResMIP. For CMIP6 and HighResMIP, the multi-model mean bias shows a too cold wedge between the surface and about 100-150 m depth extending from the west to the central-eastern tropical Atlantic. Below this and in the east, there is a warm bias. The subsurface cold bias is stronger than in HighResMIP than in CMIP6, reaching about -3°C in the model mean, which compares well to the strong subsurface cold bias seen in the ICON-O and FESOM runs shown in our manuscript. However, unlike in the HighResMIP and CMIP6 model means, in ICON-O and FESOM the subsurface cold bias extends all the way to the east, which is rather untypical. We have added this more detailed discussion of how the**

[Figure]

Figure 1: Annual mean SST bias of ICON-O spinup with respect to the HadISST dataset.

**temperature biases in ICON-O and FESOM compare to those found in CMIP6 and HighResMIP to the manuscript (see lines 260ff. and 329ff. in the revised manuscript with changes marked).**

3. Related to comment 2: How large are the biases in ICON-O and FESOM compared to those seen in typical CMIP6 models? A quick look at the AWI-CM-1-1-MR piControl simulation in the CMIP6 archive, which uses FESOM, suggests that SSTs are about 2K too cold in the western equatorial Atlantic and 2K too warm in the east. Again, it would be instructive to compare the biases in the two OGCMs with their corresponding CGCM simulations.

**As shown by Farneti et al. (2022), the SST bias at the eastern equatorial Atlantic coast is about 2 to 3°C in the CMIP6 multi-model mean and about 2°C in the HighResMIP multi-model mean. Compared to this, the eastern equatorial Atlantic SST biases are smaller in our ICON-O and FESOM runs, with about 0.5°C in ICON-O and FESOM (although you can see in the Figure given above that the eastern warm SST bias in ICON-O is rather about 1 to 1.5°C in other years than 2015, making it more similar to the CMIP6 multi-model bias). The cold SST bias in the western equatorial Atlantic is stronger in our ICON-O and FESOM runs, with -1 to -2°C, compared to about 0 in the CMIP6 and HighResMIP multi-model means (Farneti et al., 2022). The cold western subsurface equatorial temperature bias in CMIP6 is about -2°C, in HighResMIP about -3°C, in our ICON-O runs between -2 and -4°C, and in our FESOM runs between -1 and -2°C. The warm equatorial subsurface temperature bias in the east is about 2°C in the CMIP6 multi-model mean, about 1°C in HighResMIP, about 2 to 4°C in our ICON-O runs, and about 1 to 2°C in our FESOM runs (the warm eastern subsurface bias is also shifted**

and intensified to the west in ICON-O and FESOM compared to the CMIP6 and HighResMIP multi model means). We have added more details on this in the manuscript (see lines **260ff.** and **329ff.** in the revised manuscript with changes marked).

4. Prigent and Farneti (2024) have recently examined the performance of OMIP simulations in the tropical Atlantic. It should be instructive to compare with their results. Are the biases you see in your high resolution simulation similar to the OMIP simulations?
   **Figure 2 in Prigent and Farneti (2024) shows longitude-time plots of equatorial SST (3°S - 3° N). An equivalent plot for the year 2015 in our simulations and the HadISST dataset is shown in Figure 3 in this reply. In contrast to the OMIP runs in Prigent and Farneti (2024), the ICON and FESOM runs the model is too cold compared to the HadISST dataset. This holds both for the SST maximum and SST minimum. Of course, we cannot compare the plots with inter-annual variability, since the integration length for the sensitivity experiments is too short. We have added the plot shown in Figure 3 and referred to Prigent and Farneti (2024) in the section on SST in the manuscript (see Figure 7 and lines 302ff. in the revised manuscript with changes marked).**

5. SST biases in the equatorial Atlantic have a strong seasonality, with the most severe bias occurring in JJA. While the seasonality of this bias can be partly inferred from Figs. 5 and 6, it would be helpful to see a longitude-time section of the equatorial SST bias as well.
   **We agree that a longitude-time plot of the SST biases would be helpful for the reader. We have plotted the longitude-time bias in Figure 4 in this reply. The most severe bias is the cold bias, which does indeed occur in JJA. For the some of the FESOM runs, the magnitude of the warm bias is also comparable to the cold bias. We have added the plot to the manuscript (see Figure 8 and lines 312ff. in the revised manuscript with changes marked).**

6. In their conclusions the authors state that vertical mixing can only explain a limited amount of the biases seen in CGCMs, and that biases in atmosphere-ocean interaction likely play an important role. An alternative hypothesis, however, is that AGCM biases are a major source (e.g., Richter and Xie 2008, Wahl et al. 2011; Richter et al. 2012; Voldoire et al. 2018). The results shown here may be consistent with this hypothesis. The authors should discuss this.
   **We have added this to the discussion and conclusion as suggested (lines 628ff. and 662 in the revised manuscript with differences marked).**

7. What do the authors conclude about the prospect of reducing tropical Atlantic biases?
   **In light of the numerous studies that have been conducted on this topic, we conclude that the tropical Atlantic remains a challenging region for ocean models. High horizontal resolution has been suggested as a possible way to reduce biases, but we see in ICON-O and FESOM that even with 10 km horizontal resolution, biases remain large especially in the tropical Atlantic. While part of the tropical Atlantic model biases might be possible to address through tuning of ocean model parameterisations (we do see some effect of changing the vertical mixing parameters), the effect of this is highly model dependent and large biases remain even after tuning efforts. Due to the strong ocean-atmosphere coupling and feedbacks in the tropical Atlantic region, the effort to reduce the tropical Atlantic model biases has to be addressed from both the atmospheric and oceanic model components as suggested by several other studies. We have added this to our conclusions in the manuscript (lines 663ff. in the revised manuscript with differences marked).**

**2.3 Minor Comments**

1. l. 31: The study by Song et al. (2015) could also be cited here.
   **We have added the reference (line 32 in the revised manuscript).**

2. l. 99: Please define EVP.
   **We have added that EVP stands for elastic-viscous-plastic (line 102 in the revised manuscript with marked differences).**

3. Figures 7 and 8: The depth of the thermocline should be indicated in all panels.
   **We have added the depth of the thermocline in both Figures (now Figures 9 and 10 in the revised manuscript).**

4. Figure 11: The values of the vertical axes should be -0.05 instead of -0.5.
   **Thank you for pointing this out, we have changed it (now Figure 14 in the revised manuscript).**

5. Section 6: How can the bulk formula affect SST and the EUC, but not MLD?
   **The bulk formulae do affect the MLD, just not very strongly – we have changed the manuscript to describe the small change that is visible (lines 545ff. in the revised manuscript with marked changes).**

**3 RC2: Anonymous Referee**

**3.1 Summary**

The authors compare different mixing schemes in two global models to systematically address the effects of using different prescriptions of mixing on mean and seasonal variability of temperature and state variables in the tropical Atlantic. In short, while variability between models is more significant than variability between mixing schemes, the authors find that mixing schemes can affect the representation of smaller-scale phenomena; for instance only the TKE scheme reproduces diurnally-varying deep cycle turbulence. This provides more generalized insight compared to previous studies who focused on single locations with only one model. I think this work would be of interest to modelers and be a good fit in GMD.
**We would like to thank the reviewer for taking the time to provide insightful comments and are pleased that the reviewer recognises the novel contribution of our study. We are also very grateful for the attentive line-by-line comments that will certainly improve the manuscript.**

**3.2 Major comments**

1. The model resolution in the Atlantic is 10-13 km (and 50km outside the tropical Atlantic for FESOM). I agree that the resolutions in the region of interest are comparable and don't suspect that this is a problem, but I think some discussion should be added on this. That is, whether there is sufficient resolution to resolve the spatial variability in this region – While mesoscale features like tropical instability waves are probably resolved, smaller-scale filaments, etc. might not be and it is unclear how this would influence mixing.
   **It is true that the full spectrum of the sub-mescoscale is not resolved. Nevertheless, a recent study by Specht et al. (2024) demonstrate that fronts and mixing at the edges is well repesented in a 10-km ICON simulation. This suggests that the models capture at least a part of mixing and spatio-temporal variability associated with sub-mesoscale processes. We have added a short section on this at the end of the discussion section (lines 637ff. in the revised manuscript with marked changes).**

2. The current set of metrics used focuses primarily on comparing long-term mean values. While there is some comparison of seasonal variability (for SST), I think it might be important to state how well the models recreate the seasonal cycle, particularly because of strong variability associated with the equatorial current system. For example, section 4.3 discusses how well models represent the mean equatorial current, but does not include information on how well the seasonal variability is represented (even though we know from observations that it is significant). This might also provide insight into the physical mechanisms. (Or, based on the minimal influence on the seasonal cycle of SST, maybe it is similar for all runs. In any case, I think this should be discussed.)

We agree that it is helpful to see how well the models recreate the seasonal cycle of the EUC. In Figure 5 in this reply, we have plotted the monthly mean zonal velocity at 23°W averaged over the band from 3°S to 3°N as a function of depth. The choice of longitude allows for a degree of comparison with the existing plot in the paper of the EUC at 23°. The main takeaway from Figure 5 below is that both the ICON-O and FESOM simulate a distinctive seasonal cycle in terms of the maximum zonal velocity and corresponding depth. Consistent with the figure in the paper, these differences are more pronounced between the various ICON simulations than the FESOM simulations. One key difference is that zonal velocity is similar in strength in FESOM for both the MAM and SON periods, whereas in ICON the zonal velocity is clearly stronger in SON than MAM. We have added this plot to the section on the EUC in the manuscript (Figure 12 in the revised manuscript) and added some discussion on the variability of the EUC (lines 393ff. in the revised manuscript with marked changes).**

3. I think there should be some numerical quantitative results included in the text. While quantitative results can be deduced from the figures, I think that is not easy for some readers to do. I pointed out a few places that I think quantitative results would improve the text in my line-by-line comments.
**We have added some quantitative results. For details please see our responses to your line-by-line comments below.**

**3.3 Line-by-line minor comments and suggestions**

1. L1 – I would reword to not state "e.g." in the abstract. At minimum there should be a comma.
**We have reworded the sentence accordingly (line 1 in the revised manuscript).**

2. L27-33 – This section maybe can be shortened or removed... The main focus of the paper is mixing, not fixing the atmospheric or ocean parameters, right?
**While mixing is indeed the main focus, we suggest to leave this section unchanged, because it provides background on the importance of understanding contributions of bias stemming from both atmosphere and ocean models. More importantly, it highlights the fact that reducing tropical biases in coupled GCMs is not a task for ocean modellers alone.**

3. L48-56 – Move 1 paragraph earlier? – because this is a main focus of the manuscript
**We would rather leave the structure unchanged with a view to gently guide the reader through the broader motivation of our study before diving into the details of vertical turbulent mixing.**

4. L62 – delete "for example"
**We deleted this (line 63 in the revised manuscript).**

5. L65 – "specific region" instead of "specific bias"?
**We deleted this sentence and modified the following sentence to highlight that each of the aforementioned studies either focused on the effect of changing a vertical mixing parameter or the mixing scheme but not both. By contrast, Gutjahr et al. looked at both for a single model. (Lines 66ff. in the revised manuscript with marked changes)**

6. L68-69 – Strange syntax. Maybe say something like "Previous studies typically only use a single model and thus it is unclear whether those results are universally applicable"?
**We modified the sentence to improve clarity/syntax (lines 70f. in the revised manuscript with marked changes).**

7. L79 – I prefer "in the present study" to "this study". It is less ambiguous.
**We have changed "this study" to "our study" to make it clear we are referring to our study (line 81 in the revised manuscript).**

8. L83 – remove "e.g."
We have removed "e.g." and added "including", because according to a request from another reviewer, we have added some seasonal cycle SST analysis that is not limited to the Atlantic cold tongue region but covers the whole tropical Atlantic (line 85 in the revised manuscript).

9. L99 – spell out what EVP is
We have added that EVP stands for elastic-viscous-plastic (line 102 in the revised manuscript).

10. L100 – what are the boundaries for the "equatorial Atlantic" with 13 km resolution rather than 50km? That would strengthen this argument.
We added in the paper that the boundaries are at approx. 25 S and 25 N (line 103 in the revised manuscript with marked changes).

11. L101 – move this to after you discuss the ICON-O resolution.
We moved this sentence to the subsection on the ICON-O model and modified it slightly, so that it follows naturally from the previous sentence (lines 114f. in the revised manuscript with differences marked).

12. L115-116 – "we agreed on" – strange syntax, please reword
We will reword "we agreed on" and modify sentence to emphasise that we decided on the common settings before running the experiments (line 119 in the revised manuscript).

13. L122-125 – This should be elaborated on. I think the reason is because with coarser resolution, smaller-scale instances of shear instability are averaged out and thus mixing occurs at apparently higher Ri.
The bulk Richardson number is just an approximation of the gradient Richardson number for discrete model levels of finite thickness. As the levels become thinner, the bulk Richardson number becomes closer to the actual gradient Richardson number. Similarly, the critical bulk Richardson number should converge on the critical gradient Richardson number of 0.25 as the thickness of the model levels decreases. We tried to make this clearer in the text, so that there is no confusion here (lines 133ff. in the revised manuscript with marked changes).

14. L129 – Eliminate "we want to". You are making that comparison now, it is not a future research focus.
We deleted "want to" (line 142 in the revised manuscript with marked changes).

15. L131-144 – This is a very important section and provides motivation for your work. In fact, as I was reading through, I was wondering if there would be any difference if all parameters were the same in both models. Perhaps it would be a good idea to move this earlier to emphasize that point.
We agree that this is an important point and have shifted this part to the beginning of the subsection (lines 119ff. in the revised manuscript with marked changes).

16. L206+ - I'm having a hard time seeing some of the patterns in the figures that you state in the text when scrolling back and forth between the text and figures. I think labeling individual figure panels and referencing them in the text might help make this section more readable.
We agree that it is challenging to go back and forth between the text and the figures, especially with a large number of panels. However, all the panels are labelled with the corresponding experiment ID and the layout of the panels in each figure, with a column for each model, is more or less consistent throughout the paper. For this reason, we do not believe that the adding extra labels for each panel would necessarily make the life of the reader easier.

17. Another question – for some of the runs, it is clear in Figure 2 that the bias in the equatorial region is significantly different from areas to the north and south. That may be outside the domain of study, but might be something to discuss since it is so obvious in the figures.

We highlighted that the magnitude of the negative bias in the equator region is especially large (lines 237f. in the revised manuscript). This might be due to the the special equatorial dynamics, including the strong subsurface equatorial current systems, the biases of which we also discuss in the manuscript.

18. L213 – Add justification for averaging between 4N and 4S. I think this is just because of the cold tongue location.
**Yes, you are correct. We will add that the averaging area was chosen to include the cold-tongue region. We added this information (lines 238f. in the revised manuscript).**

19. L237-238 – "considerably stronger". I agree with the result itself, but in this (and other places), I think it would be insightful to include quantitative results. For example, saying 50% stronger (or however much it is) would give some better insight into the differences and I think be a more useful result.
**We agree that a quantitative comparison is useful for the reader and have added this to the text (line 267 in the revised manuscript with marked changes).**

20. L247 – ITCZ
**We corrected the typo (line 278).**

21. L245-269 (and earlier..) – Clarification question: Models typically have a warm subsurface bias in the Atlantic cold tongues. Figures 4-6 show that other than in the far eastern Atlantic, SSTs are much colder in all model runs than the observations. Would be helpful to add a sentence or two to clearly state that typical model biases are strongly depth dependent. Maybe this should go earlier in the text (line 64??)
**We added some more details on the unusual cold SST biases in our models compared to the SST biases of e.g. CMIP6 and HighResMIP (see lines 260ff. and 329ff. in the revised manuscript with changes marked). The depth dependence of the temperature biases is already extensively analysed and discussed in the manuscript.**

22. L261 – Why? I think here and in other places, as a reader I would appreciate some speculation on what is causing some of the differences between the runs.
**Part of the reason for the differences between FESOM and ICON-O are the different bulk formulae. We investigate the effect of this in Section 6 of the manuscript, and especially for the SST, which is asked about here, a large effect can be seen. We indicated this in the text at the suggested place (lines 292f. in the revised manuscript). We have only investigated this one possible factor in the ICON-O/FESOM differences because of limited computing resources, but other possibilities include e.g. differences in lateral mixing, in the horizontal grid, or in the numerical schemes and the associated numerical mixing. These speculations are already mentioned in the manuscript. We have also added this to the discussion section, as requested in your last comment below (lines 631ff. in the revised manuscript with marked changes).**

23. L276 – Remove "actually" – it sounds like you are expressing surprise, but I think we would expect similar results for both models
**We removed "actually" (line 343 in the revised manuscript with marked changes).**

24. This is another place where I think numerical, quantitative results would be helpful (yes, they can be deduced from the figures, but I think it is better if they are in the text for readers to easily see).
**We added a range of percentage differences between the runs when comparing the cold and warm biases across the models (lines 343ff. in the revised manuscript with marked changes).**

25. L283 – Mention why you specifically discuss 23W – because that's where the PIRATA mooring is
**We added this to the text (line 354 in the revised manuscript).**

26. L289-300 – Important information, but maybe this should be in the introduction?
**While we agree that the background information on EUC is important, we find that it is not necessary to move this to the introduction. This background information is intended to clarify why we are analysing the EUC in the the corresponding subsection. Although the EUC features in the introduction, the main focus is on the link between vertical mixing and tropical SST biases.**

27. L314-315 – "between 0.1 and 0.2 seems best"- Agreed, but different c_k produce more realistic results between ICON-O and FESOM. Might be helpful to state the specific values that work best in the text for each model.
**We clarified that a c_k value of 0.1 works best for ICON, whereas a c_k value of 0.2 works best for FESOM (lines 385f. in the revised manuscript).**

28. L350 – "the high stratification band is weaker" – Is this true? From Fig 10 it looks to me that it is thinner, but not necessarily weaker.
**The high stratification band is indeed weaker for FESOM than the equivalent ICON-O runs. Admittedly, it is a bit difficult to see this by eye because of the continuous color map. Since the thickness of the high-stratification band is somewhat arbitrary as it requires one to choose a lower bound of $N^2$ and it is not clear whether one can choose the same value for the ICON-O and FESOM runs, we prefer to avoid characterising the thickness of the high-stratification band.**

29. L362 – "several reasons" – If it is just those two then say that.
**We replaced "several" with "two" (line 443 in the revised manuscript with marked changes).**

30. L370-372 – True, but the atmospheric variations have a larger impact on fluxes than the diurnal jet/ocean current because wind varies more. Maybe reword this as to not overstate the impact.
**We removed "significantly" and "to a large extent", to avoid overstating impact. The section was reworded to improve clarity (lines 451ff. in the revised manuscript with marked changes).**

31. Fig 11 – I like the comparison between obs and the models. But I think the differences between model runs (e.g. discussed starting at L380) are very difficult to see other than in the top center plot. Perhaps a change in the color scale would make it easier to see these changes.
**Thank you for the suggestion regarding Fig. 11 (now Figure 14). We tested various color scales and alternative visualizations; however, we found that these approaches only marginally improved the visibility of differences between the models while slightly reducing clarity and the direct comparability with observations. For instance, if we subtract the different subplots from the observations we introduce effects due to the models' coarse temporal resolution, which diminishes the representation of the diurnal cycle. Ultimately, we chose the current visualization because the similarity between the models and observations conveys an important finding, indicating the robust reproduction of the diurnal cycle. To better highlight the differences, we provided the detailed discussion of the key discrepancies and emphasized the plot in the upper center accordingly.**

32. L381 – At different points in the text you use "parametrization", "parameterization" and "parameterization". All might be acceptable, but be consistent.
**We changed to "parameterisation" throughout the manuscript.**

33. L390 – might be worth mentioning that DCT is diurnally-varying here at the start
**We clarified that DCT is diurnally varying in the first sentence of the corresponding subsection (lines 476ff. in the revised manuscript with marked changes).**

34. L398 – Not sure I completely agree. There are periods where it looks like from Fig 12 that FESOM does not show DCT.
**We agree that the discussion was somewhat confusing and rewrote most of the text in this section (lines 484ff. in the revised manuscript with marked changes).**

35. L400-402 – I think a more needs to be added to reconcile these points. The FESOM runs seem to best represent DCT, but the ICON runs (including KPP, which poorly represents DCT) resolve the downward propagation. Aren't these related, so shouldn't the same runs represent both well? In other words, doesn't this imply that the model is reproducing DCT in the FESOM cases for the wrong reason/physics? You state later (and Fig 14 very clearly shows) that the actual K values are closer to the observations in the FESOM cases, so maybe that has something to do with it, but still I think more should be said here.
**Thank you for your comments. Indeed, the descriptions of the results were somewhat confusing. We rewrote much of the text discussing the results from the DC turbulence analysis and hope to have improved clarity of the results (lines 484ff. in the revised manuscript with marked changes).**

36. L407 – Again, it's more complicated than that. I don't think there's a single run that accurately represents all elements of DCT. So I would avoid using "satisfying" or similar terms unless you can quantify what that means.
**We removed the term "satisfying" and now emphasise that both TKE and KPP parameterisation in FESOM and ICON have advantages and disadvantages when it comes to DC turbulence (lines 511ff. in the revised manuscript with marked changes).**

37. L425 – move this to line 422? Might be better to first say what is different as a result of different bulk formulae, since I think that is the more important point
**We agree that the difference linked to the bulk formulae is the more interesting and important result. We swapped the discussion on SST and EUC with that on the MLD. (Lines 538ff. in the revised manuscript with marked changes)**

38. On another note, I think it would be helpful to explain a bit more on what terms in the bulk formula might cause these differences, referring again to appendix A.
**We are not sure if this question can be simply answered, because wind stress and heat fluxes both change by the change of the bulk formulae. To answer this question, more elaborate sensitivity experiments with isolated changes in the bulk formula terms would have to be conducted.**

39. Another question- I understand what you're doing is a sensitivity experiment, but isn't it typically unreasonable to replace bulk formula with those from another model that was tuned to different formulae? Maybe it is worth reiterating that this is a sensitivity test and I_TKE_02_FBF is not meant to get "realistic" results. (Or, I may be wrong here. In that case please clarify.)
**We added an explanation on this to the text as suggested (lines 529ff. in the revised manuscript).**

40. L442 – break up into multiple sentences
**We broke up this sentence into multiple sentences (line 562 in the revised manuscript).**

41. L477 – this sentence should be quantitative, especially considering the result is different from some past research
**We added some quantitative details as suggested (lines 605ff. in the revised manuscript with marked changes).**

42. L489 – "biases are not sensitive"
**We corrected "is" to "are" (line 618 in the revised manuscript with marked changes).**

**3.4 General comments for the discussion and the conclusion**

1. Overall I think the authors have done a nice job contextualizing their work with previous studies.
**We are happy to hear that you are satisfied with how we put our work in context in the discussion and conclusion sections.**

2. A key conclusion is that the differences between models ≫ the differences between vertical mixing schemes within models. It would be helpful to have some sort of quantitative measure of how much these differences are. This might make it easier to also compare to any previous studies,

i.e., Did the previous studies really show a greater difference between mixing schemes? Or did they show quantitatively the same effect and just perceived it as greater because they only looked at 1 model/process/variable? I think that's an important distinction to make.

**It is difficult to give quantitative results for all variables and regions. However, we have tried to add a quantitative comparison at those places in the discussion where we compare to earlier studies. The magnitude of SST and subsurface temperature changes due to the $c_k$ increase are comparable in the study by Deppenmeier et al. (2020) and in our ICON-O runs, though much smaller in FESOM (up to about 0.5°S on the surface and up to about 1.5°C in the subsurface in Deppenmeier et al. (2020), for FESOM much less, for ICON-O up to about 0.5°C on the surface and up to about 2°C in the subsurface). The values are not directly comparable because they changed their $c_k$ value from 0.1 to 0.5, whereas we only changed our $c_k$ from 0.1 to 0.3. Also, Deppenmeier et al. (2020) ran their sensitivity experiments for several decades with lower resolution, in contrast to our runs that are only two years long. Still it is interesting that the magnitude of the change compares well with that in ICON-O (though not in FESOM). Zhang et al. (2022) focus on the tropical Atlantic subsurface warm bias, which is very similar in most Ocean Model Intercomparison Project models as in ICON-O and FESOM. They show that this warm bias can be reduced in POP2 significantly by about 2.5°C by constraining the background diffusivity in the KPP scheme to observations, i.e. reducing it by one order of magnitude. Although we did not test this with the KPP scheme, we did similar runs with ICON-O using the TKE scheme. We see a similar effect as Zhang et al. (2022) describe: the subsurface warm bias is increased slightly in the ICON-O TKE run with larger background diffusivity. However, the change in the subsurface temperature bias (up to about 0.2°C) is small in our case. We added these details to the discussion (lines 578ff. in the revised manuscript with marked changes).**

3. I'd also like to see a bit of speculation on what physics are different between the models that cause the large differences. To be fair, this is touched on in a few places earlier, but I think a cohesive discussion would be useful.

   **We added a paragraph on this in the discussion section (see lines 631ff. in the revised manuscript with marked changes).**

**References**

Deppenmeier, A.-L., Haarsma, R. J., LeSager, P., and Hazeleger, W.: The effect of vertical ocean mixing on the tropical Atlantic in a coupled global climate model, Climate Dynamics, 54, 5089–5109, https://doi.org/10.1007/s00382-020-05270-x, 2020.

Farneti, R., Stiz, A., and Ssebandeke, J. B.: Improvements and persistent biases in the southeast tropical Atlantic in CMIP models, npj Climate and Atmospheric Science, 5, 42, https://doi.org/10.1038/s41612-022-00264-4, 2022.

Griffies, S. M., Danabasoglu, G., Durack, P. J., Adcroft, A. J., Balaji, V., Böning, C. W., Chassignet, E. P., Curchitser, E., Deshayes, J., Drange, H., Fox-Kemper, B., Gleckler, P. J., Gregory, J. M., Haak, H., Hallberg, R. W., Heimbach, P., Hewitt, H. T., Holland, D. M., Ilyina, T., Jungclaus, J. H., Komuro, Y., Krasting, J. P., Large, W. G., Marsland, S. J., Masina, S., McDougall, T. J., Nurser, A. J. G., Orr, J. C., Pirani, A., Qiao, F., Stouffer, R. J., Taylor, K. E., Treguier, A. M., Tsujino, H., Uotila, P., Valdivieso, M., Wang, Q., Winton, M., and Yeager, S. G.: OMIP contribution to CMIP6: experimental and diagnostic protocol for the physical component of the Ocean Model Intercomparison Project, Geoscientific Model Development, 9, 3231–3296, https://doi.org/10.5194/gmd-9-3231-2016, 2016.

Levitus, S.: Climatological atlas of the world ocean, NOAA Prof. Pap., 13, 1982.

Prigent, A. and Farneti, R.: An assessment of equatorial Atlantic interannual variability in Ocean Model Intercomparison Project (OMIP) simulations, Ocean Science, 20, 1067–1086, https://doi.org/10.5194/os-20-1067-2024, 2024.

Richter, I. and Tokinaga, H.: An overview of the performance of CMIP6 models in the tropical Atlantic: mean state, variability, and remote impacts, Climate Dynamics, 55, 2579–2601, https://doi.org/10.1007/s00382-020-05409-w, 2020.

Seo, H., Jochum, M., Murtugudde, R., Miller, A. J., and Roads, J. O.: Feedback of Tropical Instability-Wave-Induced Atmospheric Variability onto the Ocean, Journal of Climate, 20, 5842 – 5855, https://doi.org/10.1175/JCLI4330.1, 2007.

Small, R. J., Bacmeister, J., Bailey, D., Baker, A., Bishop, S., Bryan, F., Caron, J., Dennis, J., Gent, P., Hsu, H.-m., Jochum, M., Lawrence, D., Muñoz, E., diNezio, P., Scheitlin, T., Tomas, R., Tribbia, J., Tseng, Y.-h., and Vertenstein, M.: A new synoptic scale resolving global climate simulation using the Community Earth System Model, Journal of Advances in Modeling Earth Systems, 6, 1065–1094, https://doi.org/https://doi.org/10.1002/2014MS000363, 2014.

Specht, M. S., Jungclaus, J., and Bader, J.: Seasonality of Subsurface Shear Instabilities at Tropical Instability Wave Fronts in the Atlantic Ocean in a High-Resolution Simulation, Journal of Geophysical Research: Oceans, 129, e2023JC020 041, https://doi.org/10.1029/2023JC020041, 2024.

Treguier, A. M., de Boyer Montégut, C., Bozec, A., Chassignet, E. P., Fox-Kemper, B., McC. Hogg, A., Iovino, D., Kiss, A. E., Le Sommer, J., Li, Y., Lin, P., Lique, C., Liu, H., Serazin, G., Sidorenko, D., Wang, Q., Xu, X., and Yeager, S.: The mixed-layer depth in the Ocean Model Intercomparison Project (OMIP): impact of resolving mesoscale eddies, Geoscientific Model Development, 16, 3849–3872, https://doi.org/10.5194/gmd-16-3849-2023, 2023.

Zhang, Q., Zhu, Y., and Zhang, R.-H.: Subsurface Warm Biases in the Tropical Atlantic and Their Attributions to the Role of Wind Forcing and Ocean Vertical Mixing, Journal of Climate, 35, 2291–2303, https://doi.org/10.1175/jcli-d-21-0779.1, 2022.

[Figure]

Figure 2: Annual mean SST bias of FESOM run with similar setup with respect to the HadISST dataset.

[Figure]

Figure 3: Hovmoeller plot of SST averaged between 3°S and 3°N. The contours for 25 and 28°C are indicated in black.

[Figure]

Figure 4: Hovmoeller plot of SST bias relative to HadISST averaged between 3°S and 3°N.

[Figure]

Figure 5: Zonal velocity at 23°W averaged between 3°S and 3°N